# Transfer RL across Observation Feature Spaces via Model-Based Regularization

**Yanchao Sun**[†*] **Ruijie Zheng**[†] **Xiyao Wang**[†] **Andrew Cohen**[‡] **Furong Huang**[†]
[†] University of Maryland, College Park    [‡] Unity Technologies
[†]{ycs,rzheng12,xywang,furongh}@umd.edu   [‡]andrew.cohen@unity3d.com

## Abstract

In many reinforcement learning (RL) applications, the observation space is specified by human developers and restricted by physical realizations, and may thus be subject to dramatic changes over time (e.g. increased number of observable features). However, when the observation space changes, the previous policy will likely fail due to the mismatch of input features, and another policy must be trained from scratch, which is inefficient in terms of computation and sample complexity. Following theoretical insights, we propose a novel algorithm which extracts the latent-space dynamics in the source task, and transfers the dynamics model to the target task to use as a model-based regularizer. Our algorithm works for drastic changes of observation space (e.g. from vector-based observation to image-based observation), without any inter-task mapping or any prior knowledge of the target task. Empirical results show that our algorithm significantly improves the efficiency and stability of learning in the target task.

## 1 Introduction

Deep Reinforcement Learning (DRL) has the potential to be used in many large-scale applications such as robotics, gaming and automotive. In these real-life scenarios, it is an essential ability for agents to utilize the knowledge learned in past tasks to facilitate learning in unseen tasks, which is known as Transfer RL (TRL). Most existing TRL works (Taylor & Stone, 2009; Zhu et al., 2020) focus on tasks with the same state-action space but different dynamics/reward. However, these approaches do not apply to the case where the observation space changes significantly.

Observation change is common in practice as in the following scenarios. (1) Incremental environment development. RL is used to train non-player characters (NPC) in games (Juliani et al., 2018), which may be frequently updated. When there are new scenes, characters, or obstacles added to the game, the agent's observation space will change accordingly. (2) Hardware upgrade/replacement. For robots with sensory observations (Bohez et al., 2017), the observation space could change (e.g. from text to audio, from lidar to camera) as the sensor changes. (3) Restricted data access. In some RL applications (Ganesh et al., 2019), agent observation contains sensitive data (e.g. inventory) which may become unavailable in the future due to data restrictions. In these cases, the learner may have to discard the old policy and train a new policy from scratch, as the policy has a significantly different input space, even though the underlying dynamics are similar. But training an RL policy from scratch can be expensive and unstable. Therefore, there is a crucial need for a technique that transfers knowledge across tasks with similar dynamics but different observation spaces.

Besides these existing common applications, there are more benefits of across-observation transfer. For example, observations in real-world environments are usually rich and redundant, so that directly learning a policy is hard and expensive. If we can transfer knowledge from low-dimensional and informative vector observations (usually available in a simulator) to richer observations, the learning efficiency can be significantly improved. Therefore, an effective transfer learning method enables many novel and interesting applications, such as curriculum learning via observation design.

In this paper, we aim to fill the gap and propose a new algorithm that can automatically transfer knowledge from the old environment to facilitate learning in a new environment with a (drastically)

---

[*]The work was done while the author was an intern at Unity Technologies.

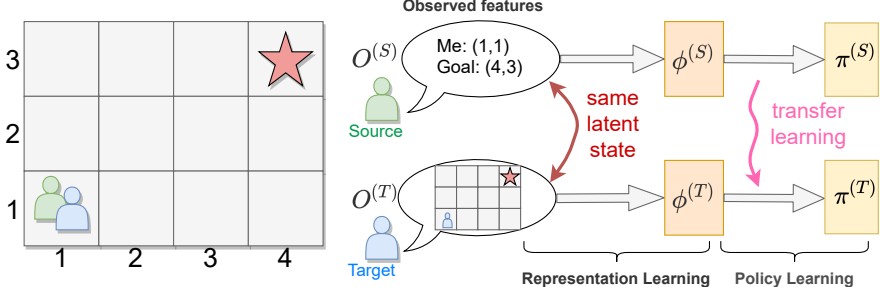

**Figure 1:** An example of the transfer problem with changed observation space. The source-task agent observes the x-y coordinates of itself and the goal, while the target-task agent observes a top-down view/image of the whole maze. The two observation spaces are drastically different, but the two tasks are structurally similar. Our goal is to transfer knowledge from the source task to accelerate learning in the target task, without knowing or learning any inter-task mapping.

different observation space. In order to meet more practical needs, we focus on the challenging setting where the observation change is: **(1) unpredictable** (there is no prior knowledge about how the observations change), **(2) drastic** (the source and target tasks have significantly different observation feature spaces, e.g., vector to image), and **(3) irretrievable** (once the change happens, it is impossible to query the source task, so that the agent can not interact with both environments simultaneously). Note that different from many prior works (Taylor et al., 2007; Mann & Choe, 2013), we do not assume the knowledge of any inter-task mapping. That is, the agent does not know which new observation feature is corresponding the which old observation feature.

To remedy the above challenges and achieve knowledge transfer, we make a key observation that, if only the observation features change, the source and target tasks share the same latent space and dynamics (e.g. in Figure 1, $\mathcal{O}^{(S)}$ and $\mathcal{O}^{(T)}$ can be associated to the same latent state). Therefore, we first disentangle representation learning from policy learning, and then accelerate the target-task agent by regularizing the representation learning process with the latent dynamics model learned in the source task. We show by theoretical analysis and empirical evaluation that the target task can be learned more efficiently with our proposed transfer learning method than from scratch.

**Summary of Contributions.** (1) To the best of our knowledge, we are the first to discuss the transfer problem where the source and target tasks have drastically different observation feature spaces, and there is no prior knowledge of an inter-task mapping. (2) We theoretically characterize what constitutes a "good representation" and analyze the sufficient conditions the representation should satisfy. (3) Theoretical analysis shows that a model-based regularizer enables efficient representation learning in the target task. Based on this, we propose a novel algorithm that automatically transfers knowledge across observation representations. (4) Experiments in 7 environments show that our proposed algorithm significantly improves the learning performance of RL agents in the target task.

## 2 PRELIMINARIES AND BACKGROUND

**Basic RL Notations.** An RL task can be modeled by a Markov Decision Process (MDP) (Puterman, 2014), defined as a tuple $M = \langle \mathcal{O}, \mathcal{A}, P, R, \gamma \rangle$, where $\mathcal{O}$ is the state/observation space, $\mathcal{A}$ is the action space, $P$ is the transition kernel, $R$ is the reward function and $\gamma$ is the discount factor. At timestep $t$, the agent observes state $o_t$, takes action $a_t$ based on its *policy* $\pi : \mathcal{O} \rightarrow \Delta(\mathcal{A})$(where $\Delta(\cdot)$ denotes the space of probability distributions), and receives reward $r_t = R(o_t, a_t)$. The environment then proceeds to the next state $o_{t+1} \sim P(\cdot|o_t, a_t)$. The goal of an RL agent is to find a policy $\pi$ in the policy space $\Pi$ with the highest cumulative reward, which is characterized by the value functions. The *value* of a policy $\pi$ for a state $o \in \mathcal{O}$ is defined as $V^\pi(o) = \mathbb{E}_{\pi, P}[\sum_{t=0}^\infty \gamma^t r_t | o_0 = o]$. The *Q value* of a policy $\pi$ for a state-action pair $(o, a) \in \mathcal{O} \times \mathcal{A}$ is defined as $Q^\pi(o, a) = \mathbb{E}_{\pi, P}[\sum_{t=0}^\infty \gamma^t r_t | o_0 = o, a_0 = a]$. Appendix A provides more background of RL.

**Representation Learning in RL.** Real-world applications usually have large observation spaces for which function approximation is needed to learn the value or the policy. However, directly learning a policy over the entire observation space could be difficult, as there is usually redundant information in the observation inputs. A common solution is to map the large-scale observation into a smaller representation space via a *non-linear encoder* (also called a *representation mapping*)

$\phi : \mathcal{O} \to \mathbb{R}^d$, where $d$ is the representation dimension, and then learn the policy/value function over the representation space $\phi(\mathcal{O})$. In DRL, the encoder and the policy/value are usually jointly learned.

# 3 PROBLEM SETUP: TRANSFER ACROSS DIFFERENT OBSERVATION SPACES

We aim to transfer knowledge learned from a source MDP to a target MDP, whose observation spaces are different while dynamics are structurally similar. Denote the source MDP as $\mathcal{M}^{(S)} = \langle \mathcal{O}^{(S)}, \mathcal{A}, P^{(S)}, R^{(S)}, \gamma \rangle$, and the target MDP as $\mathcal{M}^{(T)} = \langle \mathcal{O}^{(T)}, \mathcal{A}, P^{(T)}, R^{(T)}, \gamma \rangle$. Note that $\mathcal{O}^{(S)}$ and $\mathcal{O}^{(T)}$ can be significantly different, such as $\mathcal{O}^{(S)}$ being a low-dimensional vector space and $\mathcal{O}^{(T)}$ being a high-dimensional pixel space, which is challenging for policy transfer since the source target policy have different input shapes and would typically be very different architecturally.

In this work, as motivated in the Introduction, we focus on the setting wherein the dynamics $((P^{(S)}, R^{(S)})$ and $(P^{(T)}, R^{(T)}))$ of the two MDPs between which we transfer knowledge are defined on different observation spaces but share structural similarities. Specifically, we make the assumption that there exists a mapping between the source and target observation spaces such that the transition dynamics under the mapping in the target task share the same transition dynamics as in the source task. We formalize this in Assumption 1:

**Assumption 1.** *There exists a function* $f : \mathcal{O}^{(T)} \to \mathcal{O}^{(S)}$ *such that* $\forall o_i^{(T)}, o_j^{(T)} \in \mathcal{O}^{(T)}, \forall a \in \mathcal{A}$,

$$P^{(T)}(o_j^{(T)}|o_i^{(T)}, a) = P^{(S)}(f(o_j^{(T)})|f(o_i^{(T)}), a), \quad R^{(T)}(o_i^{(T)}, a) = R^{(S)}(f(o_i^{(T)}), a).$$

**Remarks.** (1) Assumption 1 is mild as many real-world scenarios fall under this assumption. For instance, when upgrading the cameras of a patrol robot to have higher resolutions, such a mapping $f$ can be a down-sampling function. (2) $f$ is a general function without extra restrictions. $f$ can be a many-to-one mapping, i.e., more than one target observations can be related to the same observation in the source task. $f$ can be non-surjective, i.e., there could exist source observations that do not correspond to any target observation.

Many prior works (Mann & Choe, 2013; Brys et al., 2015) have similar assumptions, but require prior knowledge of such an inter-task mapping to achieve knowledge transfer. However, such a mapping might not be available in practice. As an alternative, we propose a novel transfer algorithm in the next section that *does not* assume any prior knowledge of the mapping $f$. The proposed algorithm learns a latent representation of the observations and a dynamics model in this latent space, and then the dynamics model is transferred to speed up learning in the target task.

# 4 METHODOLOGY: TRANSFER WITH REGULARIZED REPRESENTATION

In this section, we first formally characterize *"what a good representation is for RL"* in Section 4.1, then introduce our proposed transfer algorithm based on representation regularization in Section 4.2, and next provide theoretical analysis of the algorithm in Section 4.3.

## 4.1 CHARACTERIZING CONDITIONS FOR GOOD REPRESENTATIONS

As discussed in Section 2, real-world applications usually have rich and redundant observations, where learning a good representation (Jaderberg et al., 2016; Dabney et al., 2020) is essential for efficiently finding an optimal policy. However, the properties that constitute a good representation for an RL task are still an open question. Some prior works (Bellemare et al., 2019; Dabney et al., 2020; Gelada et al., 2019) have discussed the representation quality in DRL, but we take a different perspective and focus on characterizing the sufficient properties of representation for learning a task.

Given a representation mapping $\phi$, the Q value of any $(o, a) \in \mathcal{O} \times \mathcal{A}$ can be approximately represented by a function of $\phi(o)$, i.e., $\hat{Q}(o, a) = h(\phi(o); \theta_a)$, where $h$ is a function parameterized by $\theta_a$. To study the relation between representation quality and approximation quality, we define an *approximation operator* $\mathcal{H}_\phi$, which finds the best Q-value approximation based on $\phi$. Formally, let $\Theta$ denote the parameter space of function $h \in \mathcal{H}$, then $\forall a \in \mathcal{A}, \mathcal{H}_\phi Q(o, a) := h(\phi(o); \theta_a^*)$, where $\theta_a^* = \mathrm{argmin}_{\theta \in \Theta} \mathbb{E}_o[\|h(\phi(o); \theta) - Q(\phi(o), a)\|]$. Such a function $h$ can be realized by neural networks as universal function approximators (Hornik et al., 1989). Therefore, the value approximation error $\|Q - \mathcal{H}_\phi Q\|$ only depends on the representation quality, i.e., whether we can represent the Q value of any state $o$ as a function of the encoded state $\phi(o)$.

The quality of the encoder $\phi$ is crucial for learning an accurate value function or learning a good policy. The ideal encoder $\phi$ should discard irrelevant information in the raw observation but keep essential information. In supervised or self-supervised representation learning (Chen et al., 2020; Achille & Soatto, 2018), it is believed that a good representation $\phi(X)$ of input $X$ should contain minimal information of $X$ which maintaining sufficient information for predicting the label $Y$. However, in RL, it is difficult to identify whether a representation is sufficient, since there is no label corresponding to each input. The focus of an agent is to estimate the value of each input $o \in \mathcal{O}$, which is associated with some policy. Therefore, we point out that the representation quality in RL is *policy-dependent*. Below, we formally characterize the sufficiency of a representation mapping in terms of a fixed policy and learning a task.

**Sufficiency for A Fixed Policy.** If the agent is executing a fixed policy, and its goal is to estimate the expected future return from the environment, then a representation is sufficient for the policy as long as it can encode the policy value $V_\pi$. A formal definition is provided by Definition 9 in Appendix B.

**Sufficiency for Learning A Task.** The goal of RL is to find an optimal policy. Therefore, it is not adequate for the representation to only fit one policy. Intuitively, a representation mapping is sufficient for learning if we are able to find an optimal policy over the representation space $\phi(\mathcal{O})$, which requires multiple iterations of policy evaluation and policy improvement. Definition 2 below defines a set of "important" policies for learning with $\phi(\mathcal{O})$.

**Definition 2** (Encoded Deterministic Policies). *For a given representation mapping $\phi(\cdot)$, define an encoded deterministic policy set $\Pi_\phi^D$ as the set of policies that are deterministic and take the same actions for observations with the same representations. Formally,*

$$\Pi_\phi^D := \{\pi \in \Pi \quad | \quad \exists \tilde{\pi} : \phi(\mathcal{O}) \to \mathcal{A} \text{ s.t. } \forall o \in \mathcal{O}, \pi(o) = \tilde{\pi}(\phi(o))\}, \tag{1}$$

*where $\tilde{\pi}$ is a mapping from the representation space to the action space.*

A policy $\pi$ is in $\Pi_\phi^D$ if it does not distinguish $o_1$ and $o_2$ when $\phi(o_1) = \phi(o_2)$. Therefore, $\Pi_\phi^D$ can be regarded as deterministic policies that make decisions for encoded observations. Now, we define the concept of sufficient representation for learning in an MDP.

**Definition 3** (Sufficient Representation for Learning). *A representation mapping $\phi$ is **sufficient** for a task $M$ w.r.t. approximation operator $\mathcal{H}_\phi$ if $\mathcal{H}_\phi Q_\pi = Q_\pi$ for all $\pi \in \Pi_\phi^D$. Furthermore,*

- *$\phi$ is **linearly-sufficient** for learning $M$ if $\exists \theta_a$ s.t. $Q_\pi(o, a) = \phi(o)^\top \theta_a$, $\forall a \in \mathcal{A}, \pi \in \Pi_\phi^D$.*
- *$\phi$ is **$\epsilon$-sufficient** for learning $M$ if $\|\mathcal{H}_\phi Q_\pi - Q_\pi\| \leq \epsilon$, $\forall \pi \in \Pi_\phi^D$.*

Definition 3 suggests that the representation is sufficient for learning a task as long as it is sufficient for policies in $\Pi_\phi^D$. Then, the lemma below justifies that a nearly sufficient representation can ensure that approximate policy iteration converges to a near-optimal solution. (See Appendix D for analysis on approximate value iteration.)

**Lemma 4** (Error Bound for Approximate Policy Iteration). *If $\phi$ is $\epsilon$-sufficient for task $M$ (with $\ell_\infty$ norm), then the approximated policy iteration with approximation operator $\mathcal{H}_\phi$ starting from any initial policy that is encoded by $\phi$ ($\pi_0 \in \Pi_\phi^D$) satisfies*

$$\limsup_{k \to \infty} \|Q^* - Q^{\pi_k}\|_\infty \leq \frac{2\gamma^2 \epsilon}{(1-\gamma)^2}, \tag{2}$$

*where $\pi_k$ is the policy in the $k$-th iteration.*

Lemma 4, proved in Appendix C, is extended from the error bound provided by Bertsekas & Tsitsiklis (1996). For simplicity, we consider the bound in $\ell_\infty$, but tighter bounds can be derived with other norms (Munos, 2005), although a tighter bound is not the focus of this paper.

**How Can We Learn A Sufficient Representation?** So far we have provided a principle to define whether a given representation is sufficient for learning. In DRL, the representation is learned together with the policy or value function using neural networks, but the quality of the representation may be poor (Dabney et al., 2020), which makes it hard for the agent to find an optimal policy. Based on Definition 3, a natural method to learn a good representation is to let the representation fit as many policy values as possible as auxiliary tasks, which matches the ideas in other works. For example, Bellemare et al. (2019) propose to fit a set of representative policies (called *adversarial value functions*). Dabney et al. (2020) choose to fit the values of all past policies (along the *value improvement path*), which requires less computational resource. Different from these works that directly fit

---

**Algorithm 1** Source Task Learning

---

**Require:** Regularization weight $\lambda$; update frequency $m$ for stable encoder.
1: Initialize encoder $\phi^{(S)}$, stable encoder $\hat{\phi}^{(S)}$, policy $\pi^{(S)}$, transition prediction network $\hat{P}$ and reward prediction network $\hat{R}$.
2: **for** $t = 0, 1, \cdots$ **do**
3:     Take action $a_t \sim \pi^{(S)}(\phi^{(S)}(o_t^{(S)}))$, get next observation $o_{t+1}^{(S)}$ and reward $r_t$, store to buffer.
4:     Sample a mini-batch $\{o_i, a_i, r_i, o_i'\}_{i=1}^N$ from the buffer.
5:     Update $\hat{P}$ and $\hat{R}$ using one-step gradient descent with $\nabla_{\hat{P}} L_P(\hat{\phi}^{(S)}; \hat{P})$ and $\nabla_{\hat{R}} L_R(\hat{\phi}^{(S)}; \hat{R})$, where $L_P$ and $L_R$ are defined in Equation (3).
6:     Update encoder and policy by $\min_{\pi^{(S)}, \phi^{(S)}} L_{\text{base}}(\phi^{(S)}, \pi^{(S)}) + \lambda \big( L_P(\phi^{(S)}; \hat{P}) + L_R(\phi^{(S)}; \hat{R}) \big)$.
7:     **if** $t \mid m$ **then** Update the stable encoder $\hat{\phi}^{(S)} \leftarrow \phi^{(S)}$.

---

**Algorithm 2** Target Task Learning with Transferred Dynamics Models

---

**Require:** Regularization weight $\lambda$; dynamics models $\hat{P}$ and $\hat{R}$ learned in the source task.
1: Initialize encoder $\phi^{(T)}$, policy $\pi^{(T)}$
2: **for** $t = 0, 1, \cdots$ **do**
3:     Take action $a_t \sim \pi^{(T)}(\phi^{(T)}(o_t^{(T)}))$, get next observation $o_{t+1}^{(T)}$ and reward $r_t$, store to buffer.
4:     Sample a mini-batch $\{o_i, a_i, r_i, o_i'\}_{i=1}^N$ from the buffer.
5:     Update encoder and policy by $\min_{\phi^{(T)}, \pi^{(T)}} L_{\text{base}}(\phi^{(T)}, \pi^{(T)}) + \lambda \big( L_P(\phi^{(T)}; \hat{P}) + L_R(\phi^{(T)}; \hat{R}) \big)$, where $L_P$ and $L_R$ are defined in Equation (3).

---

the value functions of multiple policies, in Section 4.2, we propose to *fit and transfer an auxiliary policy-independent dynamics model*, which is an efficient way to achieve sufficient representation for learning and knowledge transfer, as theoretically justified in Section 4.3.

## 4.2 Algorithm: Learning and Transferring Model-based Regularizer

Our goal is to use the knowledge learned in the source task to learn a good representation in the target task, such that the agent learns the target task more easily than learning from scratch. Since we focus on developing a generic transfer mechanism, the base learner can be any DRL algorithms. We use $L_{\text{base}}$ to denote the loss function of the base learner.

As motivated in Section 4.1, we propose to learn policy-independent dynamics models for producing high-quality representations: (1) $\hat{P}$ which predicts the representation of the next state based on current state representation and action, and (2) $\hat{R}$ which predicts the immediate reward based on current state representation and action. For a batch of $N$ transition samples $\{o_i, a_i, o_i', r_i\}_{i=1}^N$, define the transition loss and the reward loss as:

$$L_P(\phi, \hat{P}) = \frac{1}{N} \sum_{i=1}^N (\hat{P}(\phi(o_i), a_i) - \bar{\phi}(o_i'))^2, \quad L_R(\phi, \hat{R}) = \frac{1}{N} \sum_{i=1}^N (\hat{R}(\phi(o_i), a_i) - r_i)^2 \quad (3)$$

where $\bar{\phi}(o_i')$ denotes the representation of the next state $o_i'$ with stop gradients. In order to fit a more diverse state distribution, transition samples are drawn from an off-policy buffer, which stores shuffled past trajectories.

The learning procedures for the source task and the target task are illustrated in Algorithm 1 and Algorithm 2, respectively. Figure 2 depicts the architecture of the learning model for both source and target tasks. $z = \phi(o)$ and $z' = \bar{\phi}(o')$ are the encoded observation and next observation. Given the current encoding $z$ and the action $a$, the dynamics models $\hat{P}$ and $\hat{R}$ return the predicted next encoding $\hat{z}' = \hat{P}(z, a)$ and predicted reward $\hat{r} = \hat{R}(z, a)$. Then the transition loss is the mean squared error (MSE) between $z'$ and $\hat{z}'$ in a batch; the reward loss is the MSE between $r$ and $\hat{r}$ in a batch.

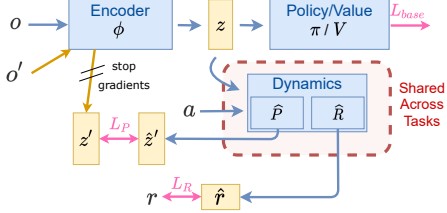

**Figure 2:** The architecture of proposed method. $\hat{P}$ and $\hat{R}$ are learned in the source task, then transferred to the target task and fixed during training.

**In the source task (Algorithm 1):** dynamics models $\hat{P}$ and $\hat{R}$ are learned by minimizing $L_P$ and $L_R$, which are computed based on a recent copy of encoder called stable encoder $\hat{\phi}^{(S)}$ (Line 5). The computation of the stable encoder is to help the dynamics models converge, as the actual encoder $\phi^{(S)}$ changes at every step. Note that a stable copy of the network is widely used in many DRL algorithms (e.g. the target network in DQN), which can be directly regarded as $\hat{\phi}^{(S)}$ without maintaining an extra network. The actual encoder $\phi^{(S)}$ is regularized by the auxiliary dynamics models $\hat{P}$ and $\hat{R}$ (Line 6).

**In the target task (Algorithm 2):** dynamics model $\hat{P}$ and $\hat{R}$ are transferred from the source task and fixed during learning. Therefore, the learning of $\phi^{(T)}$ is regularized by static dynamics models, which leads to faster and more stable convergence than naively learning an auxiliary task.

**Relation and Difference with Model-based RL and Bisimulation Metrics.** Learning a dynamics model is a common technique in model-based RL (Kipf et al., 2019; Grimm et al., 2020), whose goal is to learn an accurate world model and use the model for planning. The dynamics model could be learned on either raw observations or representations. In our framework, we also learn a dynamics model, but the model serves as an auxiliary task, and learning is still performed by the model-free base learner with $L_{\text{base}}$. Bisimulation methods (Castro, 2020; Zhang et al., 2020b) aim to approximate the bisimulation distances among states by learning dynamics models, whereas we do not explicitly measure the distance among states. Note that we also do not require a reconstruction loss that is common in literature (Lee et al., 2019).

### 4.3 THEORETICAL ANALYSIS: BENEFITS OF TRANSFERABLE DYNAMICS MODEL

The algorithms introduced in Section 4.2 consist of two designs: learning a latent dynamics model as an auxiliary task, and transferring the dynamics model to the target task. In this section, we show theoretical justifications and practical advantages of our proposed method. We aim to answer the following two questions: *(1) How does learning an auxiliary dynamics model help with representation learning? (2) Is the auxiliary dynamics model transferable?*

For notational simplicity, let $P_a$ and $R_a$ denote the transition and reward functions associated with action $a \in \mathcal{A}$. Note that $P_a$ and $R_a$ are independent of any policy. We then define the sufficiency of a representation mapping w.r.t. dynamics models as below.

**Definition 5** (Policy-independent Model Sufficiency). *For an MDP $M$, a representation mapping $\phi$ is sufficient for its dynamics $(P_a, R_a)_{a \in \mathcal{A}}$ if $\forall a \in \mathcal{A}$, there exists functions $\hat{P}_a : \mathbb{R}^d \to \mathbb{R}^d$ and $\hat{R}_a : \mathbb{R}^d \to \mathbb{R}$ such that $\forall o \in \mathcal{O}$, $\hat{P}_a(\phi(o)) = \mathbb{E}_{o' \sim P_a(o)}[\phi(o')]$, $\hat{R}_a(\phi(o)) = R_a(o)$.*

**Remarks.** (1) $\phi$ is exactly sufficient for dynamics $(P_a, R_a)_{a \in \mathcal{A}}$ when the transition function $P$ is deterministic. (2) If $P$ is stochastic, but we have $\max_{o,a} \|\mathbb{E}_{o' \sim P_a(o)}[\phi(o')] - \hat{P}_a(\phi(o))\| \leq \epsilon_P$ and $\max_{o,a} |R_a(o) - \hat{R}_a(\phi(o))| \leq \epsilon_R$, then $\phi$ is $(\epsilon_P, \epsilon_R)$**-sufficient** for the dynamics of $M$.

Next we show by Proposition 6 and Theorem 7 that learning sufficiency can be achieved via ensuring model sufficiency.

**Proposition 6** (Learning Sufficiency Induced by Policy-independent Model Sufficiency). *Consider an MDP $M$ with deterministic transition function $P$ and reward function $R$. If $\phi$ is sufficient for $(P_a, R_a)_{a \in \mathcal{A}}$, then it is sufficient (but not necessarily linearly sufficient) for learning in $M$.*

Proposition 6 shows that, if the transition is deterministic and the model errors $L_P, L_R$ are zero, then $\phi$ is exactly sufficient for learning. More generally, if the transition function $P$ is not deterministic, and model fitting is not perfect, the learned representation can still be nearly sufficient for learning as characterized by Theorem 7 below, which is extended from a variant of the value difference bound derived by Gelada et al. (2019). Proposition 6 and Theorem 7 justify that learning the latent dynamics model as an auxiliary task encourages the representation to be sufficient for learning. The model error $L_P$ and $L_R$ defined in Section 4.2 can indicate how good the representation is.

**Theorem 7.** *For an MDP $M$, if representation mapping $\phi$ is $(\epsilon_P, \epsilon_R)$-sufficient for the dynamics of $M$, then approximate policy iteration with approximation operator $\mathcal{H}_\phi$ starting from any initial policy $\pi_0 \in \Pi_\phi^D$ satisfies*

$$\limsup_{k \to \infty} \|Q^* - Q^{\pi_k}\|_\infty \leq \frac{2\gamma^2}{(1-\gamma)^3}(\epsilon_R + \gamma \epsilon_P K_{\phi,V}). \tag{4}$$

*where $K_{\phi,V}$ is an upper bound of the value Lipschitz constant as defined in Appendix B.*

**Transferring Model to Get Better Representation in Target.** Although Proposition 6 shows that learning auxiliary dynamics models benefits representation learning, finding the optimal solution is non-trivial since one still has to learn $\hat{P}$ and $\hat{R}$. Therefore, the main idea of our algorithm is to transfer the dynamics models $\hat{P}, \hat{R}$ from the source task to the target task, to ease the learning in the target task. Theorem 8 below guarantees that transferring the dynamics models is feasible. Our experimental result in Section 6 verifies that learning with transferred and fixed dynamics models outperforms learning with randomly initialized dynamics models.

**Theorem 8** (Transferable Dynamics Models). *Consider a source task $M^{(S)}$ and a target task $M^{(T)}$ with deterministic transition functions. Suppose $\phi^{(S)}$ is sufficient for $(P_a^{(S)}, R_a^{(S)})_{a \in \mathcal{A}}$ with functions $\hat{P}_a, \hat{R}_a$, then there exists a representation $\phi^{(T)}$ satisfying $\hat{P}_a(\phi(o)) = \mathbb{E}_{o' \sim P_a^{(T)}(o)}[\phi(o')]$, $\hat{R}_a(\phi(o)) = R_a^{(T)}(o)$, for all $o \in \mathcal{O}^{(T)}$, and $\phi^{(T)}$ is sufficient for learning in $M^{(T)}$.*

Theorem 8 shows that the learned latent dynamics models $\hat{P}, \hat{R}$ are transferable from the source task to the target task. For simplicity, Theorem 8 focuses on exact sufficiency as in Proposition 6, but it can be easily extended to $\epsilon$-sufficiency if combined with Theorem 7. Proofs for Proposition 6, Theorem 7 and Theorem 8 are all provided in Appendix C.

**Trade-off between Approximation Complexity and Representation Complexity.** As suggested by Proposition 6, fitting policy-independent dynamics encourages the representation to be sufficient for learning, but not necessarily linearly sufficient. Therefore, we suggest using a *non-linear policy/value head* following the representation to reduce the approximation error. Linear sufficiency can be achieved if $\phi$ is made linearly sufficient for $P_\pi$ and $R_\pi$ for all $\pi \in \Pi_\phi^D$, where $P_\pi$ and $R_\pi$ are transition and reward functions induced by policy $\pi$ (Proposition 10, Appendix B). However, using this method for transfer learning is expensive in terms of both computation and memory, as it requires to learn $P_\pi$ and $R_\pi$ for many different $\pi$'s and store these models for transferring to the target task. Therefore, there is a trade-off between approximation complexity and representation complexity. Learning a linearly sufficient representation reduces the complexity of the approximation operator. But it requires more complexity in the representation itself as it has to satisfy much more constraints. To develop a practical and efficient transfer method, we use a slightly more complex approximation operator (non-linear policy head) while keeping the auxiliary task simple and easy to transfer across tasks. Please see Appendix B for more detailed discussion about linear sufficiency.

## 5 RELATED WORK

**Transfer RL across Observation Feature Spaces.** Transferring knowledge between tasks with different observation spaces has been studied for years. Many existing approaches(Taylor et al., 2007; Mann & Choe, 2013; Brys et al., 2015) require an explicit mapping between the source and target observation spaces, which may be hard to obtain in practice. Raiman et al. (2019) introduce network surgery that deals with the change in the input features by determining which components of a neural network model should be transferred and which require retraining. However, it requires knowledge of the input feature maps, and is not designed for drastic changes, e.g. vector to pixel. Sun et al. (2020) propose a provably sample-efficient transfer learning algorithm that works for different observation spaces without knowing any inter-task mapping, but the algorithm is mainly designed for tabular RL and model-based RL which uses the model to plan for a policy, different from our setting. Gupta et al. (2017) achieve transfer learning between two different tasks by learning an invariant feature space, with a key time-based alignment assumption. We empirically compared this method with our proposed transfer algorithm in Section 6. Our work is also related to state abstraction in block MDPs, as studied by Zhang et al. (2020a). But the problem studied in Zhang et al. (2020a) is a multi-task setting where the agent aims to learn generalizable abstract states from a series of tasks. Another related topic is domain adaptation in RL (Higgins et al., 2017; Eysenbach et al., 2020; Zhang et al., 2020b), where the target observation space (e.g. real world) is different from the source observation (e.g. simulator). However, domain adaptation does not assume drastic observation changes (e.g. changed dimension). Moreover, the aim of domain adaptation is usually zero-shot generalization to new observations, thus prior knowledge or a few samples of the target domain is often needed (Eysenbach et al., 2020).

**Representation Learning in RL.** In environments with rich observations, representation learning is crucial for the efficiency of RL methods. Learning unsupervised auxiliary tasks (Jaderberg et al., 2016) is shown to be effective for learning a good representation. The relationship between learning

policy-dependent auxiliary tasks and learning good representations has been studied in some prior works (Bellemare et al., 2019; Dabney et al., 2020; Lyle et al., 2021), while our focus is to learn policy-independent auxiliary tasks to facilitate transfer learning. Using latent prediction models to regularize representation has been shown to be effective for various types of rich observations (Guo et al., 2020; Lee et al., 2019). Gelada et al. (2019) theoretically justify that learning latent dynamics model guarantees the quality of the learned representation, while we further characterize the relationship between representation and learning performance, and we utilize dynamics models to improve transfer learning. Zhang et al. (2020b) use a bisimulation metric to learn latent representations that are invariant to task-irrelevant details in observation. As pointed out by Achille & Soatto (2018), invariant and sufficient representation is indeed minimal sufficient, so it is an interesting future direction to combine our method with bisimulation metric to learn minimal sufficient representations. There is also a line of work using contrastive learning to train an encoder for pixel observations (Srinivas et al., 2020; Yarats et al., 2021; Stooke et al., 2021), which usually pre-train an encoder based on image samples using self-supervised learning. However, environment dynamics are usually not considered during pre-training. Our algorithm can be combined with these contrastive learning approaches to further improve learning performance in the target task.

## 6 EXPERIMENTAL EVALUATION

We empirically evaluate our transfer learning algorithm in various environments and multiple observation-change scenarios. Detailed experiment setup and hyperparameters are in Appendix E.

**Baselines.** To verify the effectiveness of our proposed transfer learning method, we compare our transfer learning algorithm with 4 baselines: (1) *Single*: a single-task base learner. (2) *Auxiliary*: learns auxiliary models from scratch to regularize representation. (3) *Fine-tune*: loads and freezes the source policy head, and retrains an encoder in the target task. (4) *Time-aligned* (Gupta et al., 2017): supposes the target task and the source task proceed to the same latent state given the same action sequence, and pre-trains a target-task encoder with saved source-task trajectories. More details of baseline implementations are in Appendix E.1.1.

**Scenarios.** As motivated in Section 1, there are many scenarios where one can benefit from transfer learning across observation feature spaces. We evaluate our proposed transfer algorithm in 7 environments that fit various scenarios, to simulate real-world applications:
(1) *Vec-to-pixel:* a novel and challenging scenario, where the source task has low-dimensional vector observations and the target task has pixel observations. We use 3 vector-input environments Cart-Pole, Acrobot and Cheetah-Run as source tasks, and use the rendered image in the target task.
(2) *More-sensor:* another challenging scenario where the target task has a lot more sensors than the source task. We use 3 MuJoCo environments: HalfCheetah, Hopper and Walker2d, whose original observation dimensions are 17, 11 and 17, respectively. We add mass-based inertia and velocity (provided by MuJoCo's API), resulting in 145, 91, 145 dimensions in the corresponding target tasks.
(3) *Broken-sensor:* we use an existing game 3DBall contained in the Unity ML-Agents Toolkit (Juliani et al., 2018), which has two different observation specifications that naturally fit our transfer setting: the source observation has 8 features containing the velocity of the ball; the target observation does not have the ball's velocity, thus the agent has to stack the past 9 frames to infer the velocity. Please see Appendix E.1.2 for more detailed descriptions of all the 7 environments.

**Base DRL Learners.** What we propose is a transfer learning mechanism that can be combined with any existing DRL methods. For environments with discrete action spaces (CartPole, Acrobot), we use the DQN algorithm (Mnih et al., 2015), while for environments with continuous action spaces (Cheetah-Run, HalfCheetah, Hopper, Walker2d, 3DBall), we use the SAC algorithm (Haarnoja et al., 2018). To ensure a fair comparison, we use the same base DRL learner with the same hyperparameter settings for all tested methods, as detailed in Appendix E.1.3. As is common in prior works, our implementation of the RL algorithms is mostly a proof of concept, thus many advanced training techniques are not included (e.g. Rainbow DQN).

**Results.** Experimental results on all tested environments are shown in Figure 3. We can see that our proposed transfer method learns significantly better than the single-task learner, and also outperforms all baselines in the challenging target tasks. Our transfer method outperforms Auxiliary since it transfers dynamics model from the source task instead of learning it from scratch, and outperforms Fine-tine since it regularizes the challenging encoder learning with a model-based regularizer.

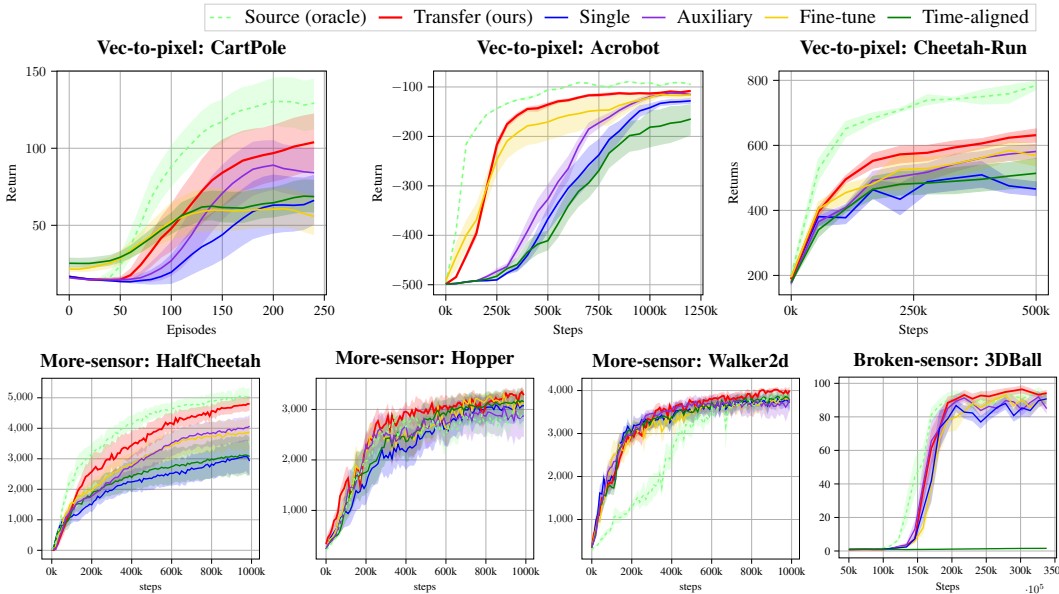

**Figure 3:** Our proposed transfer method outperforms all baselines in target tasks over all tested scenarios. (The dashed green lines are the learning curves in source tasks.) Results are averaged over 10 random seeds.

The Time-aligned method, although requires additional pre-training that is not shown in the figures, does not work better than Single in most environments, because the time-based alignment assumption may not hold as discussed in Appendix E.1.1. In some environments (e.g. Hopper, Walker2d, 3DBall), our transfer algorithm even achieves better asymptotic performance than the source-task policy, which suggests that our method can be used for improving the policy with incremental observation design. *To the best of our knowledge, we are the first to achieve effective knowledge transfer from a vector-input environment to a pixel-input environment without any pre-defined mappings.*

**Ablation Study and Hyper-parameter Test.** To verify the effectiveness of proposed transfer method, we conduct ablation study and compare our method with its two variants: only transferring the transition model $\hat{P}$ and only transferring the reward model $\hat{R}$. Figure 4 shows the comparison in HalfCheetah, and Appendix E.2 demonstrates more results. We find that all the variants of our method can make some improvements, which suggests that *transferring $\hat{P}$ and $\hat{R}$ are both effective designs for accelerating the target task learning.* Figure 6 in Appendix E.2 shows another ablation study where we investigate different selections of model regularizers and policy heads. In Algorithm 2, a hyper-parameter $\lambda$ is needed to con-

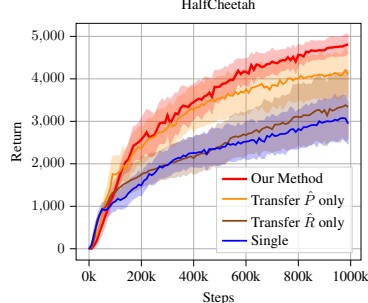

**Figure 4:** Ablation Study

trol the weight of the transferred model-based regularizer. Figure 7 in Appendix E.2 shows that, for a wide range of $\lambda$'s, the agent consistently outperforms the single-task learner.

**Potential Limitations and Solutions.** As Figure 3 shows, in some environments such as HalfCheetah, our transfer algorithm significantly outperforms baselines without transfer. But in Walker2d, the improvement is less significant, although transferring is still better than not transferring. This phenomenon is common in model-based learning (Nagabandi et al., 2018), as state predicting in Walker2d is harder than that in HalfCheetah due to the complexity of the dynamics. Therefore, we suggest using our method to transfer when the learned models $(\hat{P}, \hat{R})$ in the source task are relatively good (error is low). More techniques of improving model-based learning, such as bisimulation (Zhang et al., 2020b; Castro, 2020), can be applied to further improve the transfer performance.

## 7 CONCLUSION

In this paper, we identify and propose a solution to an important but rarely studied problem: transferring knowledge between tasks with drastically different observation spaces where inter-task mappings are not available. We propose to learn a latent dynamics model in the source task and transfer the model to the target task to facilitate representation learning. Theoretical analysis and empirical study justify the effectiveness of the proposed algorithm.

## ACKNOWLEDGEMENTS

This work is supported by Unity Technologies, National Science Foundation IIS-1850220 CRII Award 030742-00001, DOD-DARPA-Defense Advanced Research Projects Agency Guaranteeing AI Robustness against Deception (GARD), and Adobe, Capital One and JP Morgan faculty fellowships.

## ETHICS STATEMENT

Transfer learning aims to apply previously learned experience to new tasks to improve learning efficiency, which is becoming more and more important nowadays for training intelligent agents in complex systems. This paper focuses on a practical but rarely studied transfer learning scenario, where the observation feature space of an RL environment is subject to drastic changes. Driven by theoretical analysis on representation learning and its relation to latent dynamics learning, we propose a novel algorithm that transfers knowledge between tasks with totally different observation spaces, without any prior knowledge of an inter-task mapping.

This work can benefit many applications as suggested by the examples below.
(1) In many real-life environments where deep RL is used (e.g. navigating in a building), the underlying dynamics (e.g. the structure of the building) are usually fixed, but what features the observation space has is designed by human developers (e.g. what sensors are installed) and thus may change frequently during the development. When the agent gets equipped with better sensors, our algorithm makes it possible to reuse previously learned models when learning with the new sensors.
(2) An agent usually learns better with a compact observation space (e.g. a low-dimensional vector space containing its location and the goal's location) than a rich/noisy observation space (e.g. an image containing the goal). However, a compact observation is usually more difficult to construct in practice as it may require expert knowledge and human work. In this case, one can extract compact observations in a few samples and pre-train a policy with our Algorithm 1, then train the agent in the real environment with rich observations with our Algorithm 2 using the learned dynamics models. Our experiment in Figure 3 shows that the learning efficiency in the rich-observation environment can be significantly improved with our proposed transfer method.

## REPRODUCIBILITY STATEMENT

For theoretical results, we provide concrete proofs in Appendix C and Appendix D. For empirical results, we illustrate implementation details in Appendix E. The source code and running instructions are provided in the supplementary materials.

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

# Appendix: Transfer RL across Observation Feature Spaces via Model-Based Regularization

## A    ADDITIONAL PRELIMINARY KNOWLEDGE

For any policy $\pi$, its Q value $Q^\pi$ is the unique fixed point of the Bellman operator

$$(\mathcal{T}^\pi Q)(o, a) = \mathbb{E}_{o' \sim P(o, a), a' \sim \pi(o')}[R(o, a) + \gamma Q(o', a')] \tag{5}$$

The optimal policy can be found by *policy iteration* (Howard, 1960), where one starts from an initial policy $\pi_0$ and repeats policy evaluation and policy improvement. More specifically, at iteration $k$, the algorithm evaluates $Q_{\pi_k}$ via Equation (5), then improves the policy by $\pi_{k+1}(o) :=$ $\mathrm{argmax}_{a \in \mathcal{A}} Q_{\pi_k}(o, a), \forall o \in \mathcal{O}$. It is well-known that the policy iteration algorithm converges to the optimal policy under mild conditions (Puterman, 2014). When the dynamics $P$ and $R$ are unknown, reinforcement learning algorithms use interaction samples from the environment to approximately solve $\hat{Q}_{\pi_k}$. Prior works (Bertsekas & Tsitsiklis, 1996; Munos, 2005) have shown that if the approximation error is bounded by a small constant, the performance of $\pi_k$ as $k \to \infty$ is guaranteed to be close to the optimal policy value $Q_{\pi^*}$, which we also denote as $Q^*$.

## B    ADDITIONAL DISCUSSION FOR REPRESENTATION SUFFICIENCY

**Good Representation for A Fixed Policy**    We slightly abuse notation and use $\mathcal{H}_\phi$ to denote the approximation operator for state value function $V : \mathcal{O} \to \mathbb{R}$, similar to the approximation of $Q$ as introduced in Section 4.1. The following definition characterizes the sufficiency of a representation mapping in terms of evaluating a fixed policy.

**Definition 9** (Sufficient Representation for A Fixed Policy). *A representation mapping $\phi$ is **sufficient** for a policy $\pi$ w.r.t. approximation operator $\mathcal{H}_\phi$ iff $\mathcal{H}_\phi V^\pi = V^\pi$. More generally, for a constant $\epsilon \geq 0$, $\Phi$ is $\epsilon$-**sufficient** for $\pi$ iff $\|\mathcal{H}_\phi V^\pi - V^\pi\| \leq \epsilon$.*

**Remarks.** (1) If $o_1, o_2 \in \mathcal{O}$ have different values under $\pi$, a good representation should be able to distinguish them, i.e., $\phi^*(o_1) \neq \phi^*(o_2)$.
(2) The approximation operator $\mathcal{H}_\phi$ is linear if $\mathcal{H}_\phi V = \mathrm{Proj}_\Phi(V)$ where $\mathrm{Proj}_\Phi(\cdot)$ denotes the orthogonal projection to the subspace spanned by the basis functions of $\langle \phi_1, \phi_2, \cdots, \phi_d \rangle$.

**Model Sufficiency over Policies Induces Linear Learning Sufficiency**    It can be found from Definition 3 that $\phi$ is sufficient as long as it represents $Q_\pi$ for all $\pi \in \Pi_\phi^D$. Fitting various value functions to improve representation quality is proposed by some prior works Bellemare et al. (2019); Dabney et al. (2020) and shown to be effective. However, learning and fitting many policy values could be computationally expensive, and is not easy to be applied to transfer learning between tasks with different observation spaces. Can we regularize the representation with the latent dynamics instead of policy values? Proposition 10 below shows that if $\phi$ is linearly sufficient for all dynamics pairs $(P_\pi, R_\pi)$ induced by policies in $\Pi_\phi^D$ and the dynamics pairs associated with all actions, then $\phi$ is linearly sufficient for learning.

For notation simplicity, assume the state space is finite. Then, let $(P_\pi, R_\pi)$ be the transition matrix and reward vector induced by policy $\pi$, i.e., $P_\pi[i, j] = \mathbb{E}_{a \sim \pi(o_i)}[P(o_j | o_i, a)]$, $R_\pi[i] = \mathbb{E}_{a \sim \pi(o_i)}[R(o_i, a)]$. Similarly, let $P_a[i, j] = P(o_j | o_i, a)$, $R_a[i] = R(o_i, a)$. We let $\Phi$ denote the representation matrix, where the $i$-th row of $\Phi$ refers to the feature of the $i$-the observation.

**Proposition 10** (Linear Sufficiency Induced by Policy-based Model Sufficiency). *For representation $\Phi$, if for all $\pi \in \Pi_\phi^D, a \in \mathcal{A}$, there exist $\hat{P}_\pi, \hat{R}_\pi, \hat{P}_a, \hat{R}_a$ such that $\Phi \hat{P}_\pi = P_\pi \Phi$, $\Phi \hat{R}_\pi = R_\pi$, $\Phi \hat{P}_a = P_a \Phi, \Phi \hat{R}_a = R_a$, i.e. $\Phi$ is linearly sufficient both policy-based dynamics and policy-independent dynamics models, i.e., then $\Phi$ is linearly sufficient for a task $M$ w.r.t. approximation operator $\mathcal{H}_\phi$.*

Proposition 10 proven in Appendix C.3 suggests that we can let representation fit $(P_\pi, R_\pi)$ for many different $\pi$'s. However, it could be computationally intractable since the policy space is large. More

importantly, it is not memory-friendly to store and transfer a large number of dynamics models for all $(\hat{P}_\pi, \hat{R}_\pi)$. In our Proposition 6, we show that learning sufficiency can be induced by policy-independent model sufficiency, which is much simpler as there is no need to learn and store $(\hat{P}_\pi, \hat{R}_\pi)$ for many policies. As a trade-off, the policy-independent model induces non-linear sufficiency instead of linear sufficiency, requiring a more expressive approximation operator.

**Latent MDP induced by Representation**   We follow the analysis by Gelada et al. (2019) and define a new MDP under the representation mapping $\phi$: $\tilde{M} = \langle \tilde{\mathcal{O}}, \mathcal{A}, \tilde{P}, \tilde{R}, \gamma \rangle$, where for all $o \in \mathcal{O}$, $\phi(o) \in \tilde{\mathcal{O}}$, $\tilde{P}(\phi(o), a) = \hat{P}_a \phi(o)$, $\tilde{R}(\phi(o), a) = \hat{R}_a \phi(o)$. Let $\tilde{V}$ denote the value function in $\tilde{M}$, and let $\tilde{\pi}$ denote a policy in $\tilde{M}$. We make the following mild assumption

**Assumption 11.** *There exists a constant $K_{\phi,V}$ such that*
$$|\tilde{V}_{\tilde{\pi}}(\phi(o_1)) - \tilde{V}_{\tilde{\pi}}(\phi(o_2))| \leq K_{\phi,V} \|\phi(o_1) - \phi(o_2)\|, \forall \tilde{\pi} : \tilde{O} \to \mathcal{A}, o_1, o_2 \in \mathcal{O}.$$

This assumption is mild as any MDP with bounded reward has bounded value functions.

## C   Technical Proofs

### C.1   Proof of Lemma 4

*Proof of Lemma 4.* We first show that policy iteration with approximation operator $\mathcal{H}_\phi$ starting from a policy $\pi_0 \in \Pi_\phi^D$ generates a sequence of policies that are in $\Pi_\phi^D$. That is, for all $\pi_k$, and any $o_1, o_2 \in \mathcal{O}$, $\pi_k(o_1) = \pi_k(o_2)$ if $\phi(o_1) = \phi(o_2)$. We prove this claim by induction.

*Base case:* when $k = 0$, $\pi_0 \in \Pi_\phi^D$.

*Inductive step:* assume $\pi_k \in \Pi_\phi^D$ for $k \geq 0$, then for iteration $k+1$, we know that
$$\pi_{k+1}(o) := \mathrm{argmax}_a Q_k(o, a) \tag{6}$$
where $Q_k = \mathcal{H}_\phi Q_{\pi_k}$.

Based on the definition of $\mathcal{H}_\phi$, $Q_k(o, a) = f(\phi(o); \theta_a)$ for some $\theta_a$. Hence, if $o_1$ and $o_2$ have the same representation, $Q_k(o_1, \cdot)$ and $Q_k(o_2, \cdot)$ are equal. As a result, $\pi_{k+1}(o_1)$ and $\pi_{k+1}(o_2)$ are equal, so $\pi_{k+1} \in \Pi_\phi^D$.

Next we prove the error bound in Lemma 4. We start by restating the error bounds for approximate policy iteration by Bertsekas & Tsitsiklis (1996):
$$\mathrm{limsup}_{k \to \infty} \|V^* - V^{\pi_k}\|_\infty \leq \frac{2\gamma}{(1-\gamma)^2} \sup_k \|V_k - V^{\pi_k}\|_\infty \tag{7}$$

where $\pi_k$ is the policy in the $k$-th iteration. Then we extend the above result to the action value $Q$.

For any $\pi_k$ during the policy iteration (as proven above, $\pi_k \in \Pi_\phi^D$), if $\phi$ is $\epsilon$-sufficient for $M$ as defined in Definition 3 with $\ell_\infty$ norm, then we have $\|Q_k - Q_{\pi_k}\|_\infty \leq \epsilon$. That is, $\forall o \in \mathcal{O}, a \in \mathcal{A}, |Q_k(o, a) - Q_{\pi_k}(o, a)| \leq \epsilon$. Therefore, $\forall o \in \mathcal{O}$,
$$|V_k(o) - V_{\pi_k}(o)| = |\sum_{a \in \mathcal{A}} \pi(a|o)(Q_k(o, a) - Q_{\pi_k}(o, a))| \leq \epsilon \tag{8}$$

On the other hand, we can derive
$$\|Q^* - Q_{\pi_k}\|_\infty = \max_{o,a} |Q^*(o, a) - Q_{\pi_k}(o, a)| \tag{9}$$
$$= \max_{o,a} |R(o, a) + \gamma \sum_{o' \in \mathcal{O}} P(o'|o, a)V^*(o') - R(o, a) - \gamma \sum_{o' \in \mathcal{O}} P(o'|o, a)V_{\pi_k}(o')| \tag{10}$$
$$= \gamma \|V^* - V_{\pi_k}\|_\infty \tag{11}$$

Combining Equation (8) and (11), we obtain
$$\mathrm{limsup}_{k \to \infty} \|Q^* - Q_{\pi_k}\|_\infty \leq \frac{2\gamma^2}{(1-\gamma)^2} \epsilon. \tag{12}$$

$\square$

## C.2 PROOF OF PROPOSITION 6

*Proof of Proposition 6.* Given that $P$ is deterministic for $o \in \mathcal{O}, a \in \mathcal{A}$, we slightly abuse notation and let $o' = P(o, a) = P_a(o)$ if $P(o'|o, a) = 1$. If $\phi$ is sufficient for the dynamics models, i.e. $\forall o \in \mathcal{O}, a \in \mathcal{A}, \hat{P}_a \phi(o) = \phi(P_a(o)), \hat{R}_a \phi(o) = R_a(o)$. Then, we can define a new MDP $\tilde{M} = \langle \tilde{\mathcal{O}}, \mathcal{A}, \tilde{P}, \tilde{R}, \gamma \rangle$, where for all $o \in \mathcal{O}, \phi(o) \in \tilde{\mathcal{O}}$, and $\tilde{P}(\phi(o), a) = \hat{P}_a \phi(o) = \phi(P(o, a))$, $\tilde{R}(\phi(o), a) = \hat{R}_a \phi(o) = R(o, a)$.

Any policy $\pi \in \Pi_\phi^D$, based on the definition of $\Pi_\phi^D$, can be written as $\tilde{\pi} \circ \phi$, where $\tilde{\pi}$ is a deterministic policy in $\tilde{M}$. Next, we show that for all $o \in \mathcal{O}, a \in \mathcal{A}, Q_\pi(o) = \tilde{Q}_{\tilde{\pi}}(\phi(o))$.

By definition of the Q value, we know

$$Q_\pi(o, a) = \mathbb{E}_{\pi, P}[\sum_{t=0}^{\infty} \gamma^t R(o_t, a_t)|o_0 = o, a_0 = a] \tag{13}$$

$$\tilde{Q}_{\tilde{\pi}}(\phi(o)) = \mathbb{E}_{\tilde{\pi}, \tilde{P}}[\sum_{t=0}^{\infty} \gamma^t \tilde{R}(\tilde{o}_t, \tilde{a}_t)|\tilde{o}_0 = \phi(o), \tilde{a}_0 = a] \tag{14}$$

We claim that in the above equations, $\tilde{o}_t = \phi(o_t)$, $\tilde{a}_t = a_t$, for all $t \geq 0$. We prove the claim by induction.

When $t = 0$, the claim holds as $\tilde{o}_0 = \phi(o_0) = \phi(o)$, $\tilde{a}_0 = a_0 = a$.

Then, with inductive hypothesis that $\tilde{o}_t = \phi(o_t)$, $\tilde{a}_t = a_t$, we show the claim holds for $t + 1$:

Action: $a_{t+1} = \pi(o_{t+1}) = \tilde{\pi}(\phi(o_{t+1})) = \tilde{a}_{t+1}$.

State: $\tilde{o}_{t+1} = \tilde{P}(\tilde{o}_t, a_t) = \tilde{P}(\phi(o_t), a_t) = \phi(P(o_t, a_t)) = \phi(o_{t+1})$.

Hence, we have shown $\tilde{o}_t = \phi(o_t)$, $\tilde{a}_t = a_t$, for all $t \geq 0$, then the reward in the $t$-th step of Equation (13) and (14) are the same, as $\tilde{R}(\tilde{o}_t, \tilde{a}_t) = \tilde{R}(\phi(o_t), a_t) = R(o_t, a_t)$. Therefore, $Q_\pi(o) = \tilde{Q}_{\tilde{\pi}}(\phi(o))$.

Therefore, for any $\pi \in \Pi_\phi^D$, its action value can be represented by $\tilde{Q}_{\tilde{\pi}} \circ \phi$. Therefore, $\phi$ is sufficient for learning in $M$.

Next, we show that $\phi$ **is not necessarily linearly sufficient** for learning the task.

Consider an arbitrary policy $\pi \in \Pi_\phi^D$. Without loss of generality, suppose $R_\pi$ is linearly represented by $\phi$, i.e. $R_a(o, a) = \phi(o)^\top \hat{R}_a$, then we have

$$R_\pi(o) = \sum_{a \in \mathcal{A}} \pi(a|\phi(o))R_a(o, a)$$
$$= \sum_{a \in \mathcal{A}} \pi(a|\phi(o))\phi(o)^\top \hat{R}_a$$
$$= \langle \pi(\phi(o)), \hat{R}^\top \phi(o) \rangle$$
$$= \langle \hat{R}\pi(\phi(o)), \phi(o) \rangle$$

where $\hat{R} := [\hat{R}_{a_1}; \hat{R}_{a_1}; \cdots; \hat{R}_{a_{|\mathcal{A}|}}]$. We can find that unless $\pi$ always takes the same action for all input states, $\phi$ is not guaranteed to linearly encode $R_\pi$.

Similarly, for $P_\pi$, suppose $P_a(\cdot|o, a) = \phi(o)^\top \hat{P}_a$ we have

$$\phi(P_\pi(o)) = \sum_{a \in \mathcal{A}} \pi(a|\phi(o))P_a(\cdot|o, a)$$
$$= \sum_{a \in \mathcal{A}} \pi(a|\phi(o))\phi(o)^\top \hat{P}_a$$
$$= \hat{P}(\pi(\phi(o)), \phi(o), I)$$

where $\hat{P} := [\hat{P}_{a_1}; \hat{P}_{a_1}; \cdots; \hat{P}_{a_{|\mathcal{A}|}}]$ is an $|\mathcal{A}| \times d \times d$ tensor, and $\hat{P}(\cdot, \cdot, \cdot)$ denotes the multi-linear operation. Hence, if $\pi$ takes different actions in different states, $\phi$ may not linearly encode the transition, either.

Therefore, $\phi$ is not guaranteed to linearly encode $R_\pi$ and $P_\pi$, and thus is not guaranteed to linearly encode $V_\pi$ and $Q_\pi$.

□

## C.3 PROOF OF PROPOSITION 10

*Proof of Proposition 10.* We first show that for any $\pi \in \Pi$, if $\phi$ is linearly sufficient for $(P_\pi, R_\pi)$, then there exists a vector $\omega \in \mathbb{R}^k$ such that $V_\pi = \hat{V}_\pi = \Phi\omega$.

Since $\Phi$ is linearly sufficient for $P_\pi$ and $R_\pi$, we have $\Phi\hat{P}_\pi = P_\pi\Phi$ and $\Phi\hat{R}_\pi = R_\pi$ for some $\hat{P}_\pi$ and $\hat{R}_\pi$. Let $\omega = (I - \gamma\hat{P}_\pi)^\dagger \hat{R}_\pi$, then the Bellman error of $\hat{V}_\pi = \Phi\omega$ can be computed as

$$
\begin{aligned}
R_\pi + \gamma P_\pi \hat{V}_\pi - \hat{V}_\pi &= R_\pi + \gamma P_\pi \Phi\omega - \Phi\omega \\
&= \Phi\hat{R}_\pi + \gamma\Phi\hat{P}_\pi\omega - \Phi\omega \\
&= \Phi(\hat{R}_\pi - (I - \gamma\hat{P}_\pi)(I - \gamma\hat{P}_\pi)^\dagger \hat{R}_\pi) \\
&= \Phi(\hat{R}_\pi - \hat{R}_\pi) \\
&= 0
\end{aligned}
$$

Therefore, $\hat{V}_\pi$ is a fixed point of the Bellman operator $\mathcal{T}^\pi$, and thus equal to $V_\pi$.

Next, as we know that $Q_\pi(\cdot, a) = R_a + \gamma\langle P_a, V_\pi\rangle$, and $\Phi\hat{P}_a = P_a\Phi, \Phi\hat{R}_a = R_a$, we can obtain

$$
\begin{aligned}
Q_\pi(\cdot, a) = R_a + \gamma\langle P_a, V_\pi\rangle \\
&= \Phi\hat{R}_a + \gamma P_a\Phi\omega \\
&= \Phi\hat{R}_a + \gamma\Phi\hat{P}_a\omega \\
&= \Phi(\hat{R}_a + \gamma\hat{P}_a\omega)
\end{aligned}
$$

Therefore, for any $\pi \in \Pi_\phi^D$, $Q_\pi$ can be linearly represented by $\Phi$, and thus $\Phi$ is linearly sufficient for learning by definition.

□

## C.4 PROOF OF THEOREM 7

*Proof of Theorem 7.* Lemma 2 in Gelada et al. (2019) is based on one policy in the induced MDP and bounded model errors. We can replace the Wasserstein distance $\mathcal{W}(\phi P(\cdot|o, a), \tilde{P}(\cdot|\phi(o), a))$ by the Euclidean distance $\|\phi P(o, a), \tilde{P}(\phi(o), a)\|$ as we focus on deterministic transitions.

For any policy $\pi \in \Pi_\phi^D$ that can be written as $\tilde{\pi} \circ \phi$, we have

$$
|Q_\pi(o, a) - \tilde{Q}_{\tilde{\pi}}(\phi(o), a)| \leq \frac{\epsilon_R + \gamma K_{\phi,V}\epsilon_P}{1 - \gamma} \tag{15}
$$

Therefore, $\phi$ is $(1 - \gamma)^{-1}(\epsilon_R + \gamma K_{\phi,V}\epsilon_P)$-sufficient for learning $M$.

Combined with Lemma 4, we can obtain the bound in Theorem 7.

□

## C.5 PROOF OF THEOREM 8

*Proof of Theorem 8.* First of all, if there exists $\phi^{(T)}$ satisfying $\hat{P}_a(\phi(o_i)) = P_a^{(T)}[i]\Phi^{(T)}, \hat{R}_a(\phi(o_i)) = R_a^{(T)}[i], \forall o_i \in \mathcal{O}^{(T)}$, then it is sufficient for the dynamics of the target task, and thus sufficient for

learning the target task as stated in Proposition 6. Therefore, our focus is to show the existence of such a representation.

As $\phi^{(S)}$ is sufficient for $P_a^{(S)}$ and $R_a^{(S)}$ for all $a \in \mathcal{A}$, we have

$$\hat{P}_a(\phi^{(S)}(o^{(S)})) = P^{(S)}(o^{(S)}, a)\Phi^{(S)} = \phi^{(S)}(P^{(S)}(o^{(S)}, a)) \tag{16}$$

$$\hat{R}_a(\phi^{(S)}(o^{(S)})) = R^{(S)}(o^{(S)}, a) \tag{17}$$

where we let $P^{(S)}(o^{(S)}, a)$ denote the next state of $(o^{(S)}, a)$, given that $P$ is deterministic.

Based on Assumption 1, we know that there exists a function $f$ such that $\forall o^{(T)} \in \mathcal{O}^{(T)}$, $f(o^{(T)}) \in \mathcal{O}^{(S)}$, and $f(P^{(T)}(o^{(T)}, a)) = P^{(S)}(f(o^{(T)}), a)$, $R^{(T)}(o^{(T)}, a) = R^{(S)}(f(o^{(T)}), a)$. Hence, we can obtain

$$\hat{P}_a(\phi^{(S)}(f(o^{(T)}))) = \phi^{(S)}(P^{(S)}(f(o^{(T)}), a)) = \phi^{(S)}(f(P^{(T)}(o^{(T)}, a))) \tag{18}$$

$$\hat{R}_a(\phi^{(S)}(f(o^{(T)}))) = R^{(S)}(f(o^{(T)}), a) = R^{(T)}(o^{(T)}, a) \tag{19}$$

Let $\hat{\phi}^{(T)} := \phi^{(S)} \circ f$, then we get

$$\hat{P}_a(\hat{\phi}^{(T)}(o^{(T)})) = \hat{\phi}^{(T)}(P^{(T)}(o^{(T)}, a)) \tag{20}$$

$$\hat{R}_a(\hat{\phi}^{(T)}(o^{(T)})) = R^{(T)}(o^{(T)}, a) \tag{21}$$

Therefore, $\hat{\phi}^{(T)}$ is a feasible solution satisfying model sufficiency in the target task, and thus is sufficient for learning.

Theorem 8 holds since we have shown (1) all feasible solutions to $\Phi^{(T)}\hat{P}_a = P_a^{(T)}\Phi$ and $\Phi^{(T)}\hat{R}_a = R_a^{(T)}$ are sufficient for learning in $M^{(T)}$, and (2) there exists at least one feasible solution to $\Phi^{(T)}\hat{P}_a = P_a^{(T)}\Phi$ and $\Phi^{(T)}\hat{R}_a = R_a^{(T)}$.

$\square$

## D REPRESENTATION LEARNING FOR APPROXIMATE VALUE ITERATION

Now we illustrate how our transfer algorithm and the proposed model-based regularization work for approximate value iteration. We focus on the case where the reward function $R(o, a) \geq 0$ for all $o \in \mathcal{O}$ and $a \in \mathcal{A}$.

**Preliminaries**  The bases of value iteration is the Bellman optimality operator $\mathcal{T}^*$. For the value function, we have

$$\mathcal{T}^*V(o) = \max_{a \in \mathcal{A}}[R(o, a) + \gamma \sum_{o \in \mathcal{O}} P(o'|o, a)V(o')] \tag{22}$$

For the Q function, we have

$$\mathcal{T}^*Q(o, a) = R(o, a) + \gamma \sum_{o \in \mathcal{O}} P(o'|o, a)\max_{a' \in \mathcal{A}} Q(o', a'), \tag{23}$$

where we slightly abuse notation and use $\mathcal{T}^*$ for both the value function and the Q function when there is no ambiguity.

Starting from some initial $Q_0$ (or $V_0$) and iteratively applying $\mathcal{T}^*$, i.e., $Q_{k+1} = \mathcal{T}^*Q_k$, $Q_k$ (or $V_k$) can finally converge to $Q^*$ or $V^*$ when $k \to \infty$, which is known as value iteration. When an approximation operator $\mathcal{H}$ is used, the process is called approximate value iteration (AVI): $Q_{k+1} = \mathcal{H}_\phi \mathcal{T}^*Q_k$. A prior work has shown the following asymptotic result:

**Lemma 12** (Approximate Value Iteration for $Q$). *For a reinforcement learning algorithm based on value iteration with approximation operator $\mathcal{H}$, if $\|\mathcal{H}\mathcal{T}^*Q_k - \mathcal{T}^*Q_k\|_\infty \leq \epsilon$, for all $Q_k$ along the value iteration path, then we have*

$$\limsup_{k \to \infty} \|V^* - V_{\pi_k}\| \leq \frac{2\epsilon}{(1 - \gamma)^2}. \tag{24}$$

**Value Iteration Learning with Given Representation**  Given a representation mapping $\phi$, we aim to learn an approximation function $h : \phi(\mathcal{O}) \to \mathbb{R}^{|A|}$ such that $\hat{Q}(o, \cdot) = h(\phi(o)) \approx Q(o, \cdot)$. For notation simplicity, we further use $h(\phi(o), a)$ to denote the approximated Q value $\hat{Q}(o, a)$ for $a \in \mathcal{A}$. We start from an initial $h_0$ that has a uniform value $c$ for all inputs, where $c > 0$ can be randomly selected. The initial approximation for Q value is $\hat{Q}_0 = h_0 \circ \phi$. Then, at iteration

$k > 0$, we solve $\hat{Q}_k = \mathcal{H}_\phi \mathcal{T}^* \hat{Q}_{k-1} = h_k \circ \phi$, where $h_k := \arg\min_h \|h \circ \phi - \mathcal{T}^* \hat{Q}_{k-1}\|_\infty$. We use a neural network (universal function approximator) to parameterize $h$, so the approximation error $\|h \circ \phi - \mathcal{T}^* \hat{Q}_{k-1}\|_\infty$ depends on the representation quality of $\phi$.

Therefore, a representation mapping $\phi$ is $\epsilon$-sufficient for learning with value iteration if $\|\mathcal{H}_\phi \mathcal{T}^* Q_k - \mathcal{T}^* Q_k\|_\infty \leq \epsilon$. Next, we identify the relationship between policy-independent model sufficiency and the learning sufficiency with value iteration methods.

**Guaranteed Learning with Model-regularized Representation**    We first make the following assumption for the learned approximation function $h$.

**Assumption 13** (Lipschitz Value Approximation). *There exists a constant $K_{\phi,h}$, such that $\forall k \geq 0, o_1, o_2 \in \mathcal{O}, a \in \mathcal{A}$,*

$$|h_k(\phi(o_1), a) - h_k(\phi(o_2), a)| \leq K_{\phi,h} \|\phi(o_1) - \phi(o_2)\|, \tag{25}$$

*where $h_k$ is the approximation function in the $k$-th iteration.*

Then, the following Theorem holds, which justifies that learning with model-regularized representation helps with value iteration learning.

**Theorem 14.** *For an MDP $M$, if encoder $\phi$ satisfies $\max_{o \in \mathcal{O}, a \in \mathcal{A}} |R(o, a) - \hat{R}_a(\phi(o))| \leq \epsilon_R$ and $\max_{o \in \mathcal{O}, a \in \mathcal{A}} \|\mathbb{E}_{o' \sim P(\cdot|o,a)} \phi(o') - \hat{P}_a(\phi(o))\|_2 \leq \epsilon_P$ for dynamics models $(\hat{P}_a, \hat{R}_a)_{a \in \mathcal{A}}$, then the approximated value iteration with approximation operator $\mathcal{H}_\phi$ under Assumption 13 satisfies*

$$\limsup_{k \to \infty} \|V^* - V^{\pi_k}\|_\infty \leq \frac{2}{(1-\gamma)^2} (\epsilon_R + \gamma \epsilon_P K_{\phi,h}). \tag{26}$$

*Proof of Theorem 14.* Let $h_k$ be the approximation function in the $k$-th iteration. That is, the approximated Q function in the $k$-th iteration is $\hat{Q}_k = h_k \circ \phi$. As the rewards of all state-action pairs are non-negative, we have $h_k(\phi(o), a) \geq 0$.

Define a function $\hat{h}_{k+1}$ as

$$\hat{h}_{k+1}(\phi(o), a) = \hat{R}_a(\phi(o)) + \gamma \max_{a' \in \mathcal{A}} h_k(\hat{P}_a(\phi(o)), a') \tag{27}$$

Given that

$$\mathcal{T}^* \hat{Q}_k(o, a) = R(o, a) + \gamma \mathbb{E}_{o' \sim P(\cdot|o,a)} [\max_{a' \in \mathcal{A}} \hat{Q}_k(o', a')] \tag{28}$$

$$= R(o, a) + \gamma \mathbb{E}_{o' \sim P(\cdot|o,a)} [\max_{a' \in \mathcal{A}} h_k(\phi(o'), a')], \tag{29}$$

we have that for any $o \in \mathcal{O}, a \in \mathcal{A}$,

$$\left| \hat{h}_{k+1}(\phi(o), a) - \mathcal{T}^* \hat{Q}_k(o, a) \right| \tag{30}$$

$$= \left| \hat{R}_a(\phi(o)) + \gamma \max_{a' \in \mathcal{A}} h_k(\hat{P}_a(\phi(o)), a') - (R(o, a) + \gamma \mathbb{E}_{o' \sim P(\cdot|o,a)} [\max_{a' \in \mathcal{A}} h_k(\phi(o'), a')]) \right| \tag{31}$$

$$\leq \left| \hat{R}_a(\phi(o)) - R(o, a) \right| + \gamma \left| \max_{a' \in \mathcal{A}} h_k(\hat{P}_a(\phi(o)), a') - \mathbb{E}_{o' \sim P(\cdot|o,a)} [\max_{a' \in \mathcal{A}} h_k(\phi(o'), a')] \right| \tag{32}$$

$$\leq \epsilon_R + \gamma \mathbb{E}_{o' \sim P(\cdot|o,a)} \left[ \left| \max_{a' \in \mathcal{A}} h_k(\hat{P}_a(\phi(o)), a') - \max_{a' \in \mathcal{A}} h_k(\phi(o'), a') \right| \right] \tag{33}$$

$$\leq \epsilon_R + \gamma \mathbb{E}_{o' \sim P(\cdot|o,a)} \max_{a' \in \mathcal{A}} \left| h_k(\hat{P}_a(\phi(o)), a') - h_k(\phi(o'), a') \right| \tag{34}$$

$$\leq \epsilon_R + \gamma \mathbb{E}_{o' \sim P(\cdot|o,a)} \max_{a' \in \mathcal{A}} K_{\phi,h} \left\| \hat{P}_a(\phi(o) - \phi(o')) \right\| \tag{35}$$

$$\leq \epsilon_R + \gamma \epsilon_P K_{\phi,h}, \tag{36}$$

where (34) is due to the non-negativity of $h_k$, and (35) is due to Assumption 13.

Now we have shown that the constructed $\hat{h}_{k+1}$ satisfies

$$\|\hat{h}_{k+1} \circ \phi - \mathcal{T}^* \hat{Q}_k\|_\infty \leq \epsilon_R + \gamma \epsilon_P K_{\phi,h}. \tag{37}$$

According to the definition of $\mathcal{H}_\phi$, we obtain

$$\|h_{k+1} \circ \phi - \mathcal{T}^* \hat{Q}_k\|_\infty \leq \|\hat{h}_{k+1} \circ \phi - \mathcal{T}^* \hat{Q}_k\|_\infty \leq \epsilon_R + \gamma \epsilon_P K_{\phi,h} \tag{38}$$

since $\mathcal{H}_\phi$ finds a $h_{k+1}$ that minimizes the approximation error.

Therefore, Theorem 14 follows by combining Inequality (38) and Lemma 12.

$\square$

# E  EXPERIMENT DETAILS AND ADDITIONAL RESULTS

## E.1  EXPERIMENT SETTING DETAILS

### E.1.1  BASELINES

- **Single**: A DQN or SAC learner on the target domain without any auxiliary tasks.
- **Auxiliary**: On the target domain, the encoder $\phi^{(T)}$ is optimized based on the loss $L_{\text{base}}(\phi^{(T)}, \pi^{(T)}) + \lambda\left[L_P(\phi^{(T)}; \hat{P}^{(T)}) + L_R(\phi^{(T)}; \hat{R}^{(T)})\right]$. Compared with our transfer algorithms which transfer the learned dynamics from source domain to the target domain, it learns the dynamics model $(\hat{P}^{(T)}, \hat{R}^{(T)})$ on the target domain from scratch. Here we set $\lambda$ to be the same as our transferred algorithm (values of $\lambda$ are provided in Appendix E.1.3). The purpose of this baseline is to test whether the efficiency of our proposed transfer algorithms come from the transferred latent dynamics or from the auxiliary loss (or potentially both).
- **Fine-tune**: To test whether our transfer algorithms benefit from loading the learned policy head $\pi^{(S)}$, on the target domain, we load the weights of $\pi^{(T)}$ from the trained source policy head $\pi^{(S)}$ and train the DQN or SAC agent without any auxiliary loss.
- **Time-aligned**: Gupta et al. (2017) propose to learn aligned representations for two tasks, under the assumption that the source-task agent and the target-task agent reach similar latent sates at the same time step, i.e. $\phi^{(T)}(s_t^{(T)}) = \phi^{(S)}(s_t^{(S)})$. Note that this assumption is valid when the initial state is fixed and the transitions are all deterministic. Although in our setting, the agent can not learn both tasks simultaneously, we can adapt the idea of time-based alignment and encourage the target encoder to map target observations to the source representations happening at the same time step.

  In our experiments, we store $N$ source trajectories $\left\{s_0^i, a_0^i, s_1^i, a_1^i, ...,\right\}_{i=1}^N$ collected during source task training. Then on the target domain, we first collect $N$ trajectories following the same action as the one collected from the source domain. In other words, at time step $t$ of the $i$-th trajectory, we take action $a_t^i$. After the target trajectories are collected, we minimize the alignment loss $L_{\text{align}}(\phi^{(T)}) = \mathbb{E}\left[\left(\phi^{(T)}(s_t^{(T)}) - \phi^{(S)}(s_t^{(S)})\right)^2\right]$ to enforce that observations from source and target domain at the same time-step have the same representations.

  In our experiments, we set $N$ to be 10% of the training trajectories. (We also experimented with larger $N$', for example using all the training trajectories, but the differences are minor.) In terms the alignment loss, we optimize the loss for 1000 epochs with batch size equal to 256, where at each epoch we sample a batch of paired source and target observations and compute the alignment loss. After pre-training the target encoder, we load the weight into $\phi^{(T)}$ and resumes the normal DQN or SAC training.

  Our experimental results show that, although more training steps are given to the time-aligned learner, it does not outperform the single-task learner, and sometimes fails to learn (e.g. in 3DBall). The main reason is that the time-based assumption does not hold in practice as initial states are usually randomly generated. Therefore, even though the agent exactly imitates the source-task policy at every step, the observations from source and target task do not necessarily match at every time-step. In environments with non-deterministic transitions, the state mismatch will be a more severe issue and may lead to an unreasonable encoder.

### E.1.2  ENVIRONMENTS

**Environment Settings in Vec-to-pixel Tasks**

- CartPole: The source task is the same as the ordinary CartPole environment on Gym. For the pixel-input target task, we extract the screen of the environment which is of size

(400,600), and crop the pixel input to let the image be centered at the cart. The resulting observation has size (40,90) after cropping. We take the difference between two consecutive frames as the agent's observation.

- Acrobot: The source task is the same as the ordinary Acrobot environment on Gym. For the pixel-input target task, we first extract the screen of the environment which is of size (150,150), and then down-sample the image to (40,40). We also take the difference between two consecutive frames as the agent's observation.

- Cheetah-Run: The source task is the Cheetah Run Task provided by DeepMind Control Suite (DMC) (Tassa et al., 2018). For the target task, we use the image of size (84,84) rendered from the environment as the agent's observation.

**Environment Settings in More-sensor Tasks**   For the target task of MuJoCo environments, we add the center of the mass based inertia and velocity into the observations of the agent, concatenating them with the original observation on the source task. Consequently, in the target environments, the dimensionality of the observation space on target task become much larger than that of the source task. On Hopper, the dimensionality of the target observation is 91, whereas the the source observation space only has 11 dimensions. The dimensionalities of target tasks on HalfCheetah, Hopper and Walker are 145, 91, 145 respectively.

**Environment Settings in Broken-sensor Tasks**   3DBall is an example environment provided by the ML-Agents Toolkit (Juliani et al., 2018). In this task, the agent (a cube) is supposed to balance a ball on its top. At every step, the agent will be rewarded if the ball is still on its top. If the ball falls off, the episode will immediately end. The highest episodic return in this task is 100. There are two versions of this game, which only differ by their observation spaces. The simpler version (named 3DBall in the toolkit) has 8 observation features corresponding to the rotation of the agent cube, and the position and velocity of the ball. The harder version (named 3DBallHard in the toolkit) does not have access to the ball velocity, but observes a stack of 9 past frames, each of which corresponds to the rotation of the agent cube, and the position of the ball, resulting in 45 observation dimensions at every step. We regard 3DBall as the source task and 3DBallHard as the target task in our experiments.

### E.1.3   IMPLEMENTATION OF BASE DRL ALGORITHMS AND HYPER-PARAMETER SETTINGS

**Implementation of DQN**   To ensure that the base learning algorithm learns the pixel-input target tasks well, we follow the existing online codebases for pre-processing, architectures and hyperparameter settings in pixel CartPole[1] and pixel Acrobot [2]. On source domain, the DQN network has a 2-layer encoder and a 2-layer Q head of hidden size 64, and the representation dimension is set as 16. For pixel-input, the encoder has three convolution layers followed by a linear layer. The number of channels of the convolutional layers are equal to 16, 32, 32, respectively (kernel size=5 for all three layers). We use the Adam optimizer with learning rate $0.001$ and $\beta_1, \beta_2 = 0.9, 0.999$. The target Q network is updated every 10 iterations. In CartPole, we use a replay buffer with size 10000. In the more challenging Acrobot, we use a prioritized replay buffer with size 100000.

**Implementation of SAC**   For MuJoCo environments, we follow an elegant open-sourced SAC implementation[3]. The number of hidden units for all neural networks is 256. The actor has a two-layer encoder and a two-layer policy head. The two Q networks both have three linear layers. The activation function is ReLU and the learning rate is $3 \cdot 10^{-4}$. We train the dynamics model and the reward model every 50k interactive steps in the source task. For the DMC environment Cheetah-Run, we follow the open-sourced SAC implementation with an autoencoder [4]. The pixel encoder has three convolution layers and one linear layer. The number of channels for all convolutional layers is 32 and the kernel size is 3. For the 3DBall environment, as it can only be learned within the ML-Agents toolkit, we directly use the SAC implementation provided by the toolkit with the default hyperparameter settings.

---

[1]https://pytorch.org/tutorials/intermediate/reinforcement_q_learning.html
[2]https://github.comeyalbd2Deep_RL_Course
[3]https://github.com/pranz24/pytorch-soft-actor-critic
[4]https://github.com/denisyarats/pytorch_sac_ae

**Implementation of Latent Dynamics Model**   Note that our goal is to learn a good representation by enforcing it predicting the latent dynamics, different from model-based RL (Hafner et al., 2019) that aims to learn accurate models for planning. Therefore, we let the dynamics models $\hat{P}$ and $\hat{R}$ be simple linear networks, so that the representation can be more informative in terms of representing dynamics and learning values/policies. For environments with discrete action spaces, we learn $|\mathcal{A}|$ linear transition networks and $|\mathcal{A}|$ linear reward models. For environments with continuous action spaces, we first learn an action encoder $\psi : \mathcal{A} \to \mathbb{R}^d$ with the same encoding size $d$ as the state representation. Then, we learn a linear transition network and a linear reward network with $\hat{P}(\phi(o) \circ \psi(a))$ being the predicted next representation, and $\hat{R}(\phi(o) \circ \psi(a))$ being the predicted reward, where $\circ$ denotes element-wise product. In practice, we find this implementation achieves good performance across many environments.

In addition, note that due to the significant difference between source observation and target observation, the initial encoding scale could be very different in source and target tasks, making it hard for them to be regularized by the same dynamics model. Therefore, we normalize the output of both encoders to be a unit vector (l2 norm is 1), which remedies the potential mismatch in their scales.

**Hyperparameter Settings for Transfer Learning**   In experiments, we find that it is better to set $\lambda$ relatively large when the environment dynamics are simple and the dynamics model is of high quality. When the environment dynamics is complex, we choose to be more conservative and set $\lambda$ to be smaller. Concretely, in CartPole, $\lambda$ is set as 18; in 3DBall, $\lambda$ is set as 10; in Acrobot, $\lambda$ is set as 5; in the remaining MuJoCo environments where dynamics are more complicated, $\lambda$ is set as 1. Although we use different $\lambda$'s in different environments based on domain knowledge, we find that different values of $\lambda$'s do not have much influence on the learning performance. Figure 7 provided in Appendix E.2 shows a test on the hyper-parameter $\lambda$, where we can see that our algorithm effectively transfers knowledge under various values of $\lambda$.

Regarding the representation dimension, we set it to be smaller for simpler tasks, and larger for more complex tasks. In 3DBall, we set the encoding size to be 8; in CartPole, we set the encoding size as 16; in Acrobot, we set the encoding size as 32; in Cheetah-Run, we set the encoding size as 50; in MuJoCo tasks, we set the encoding size as 256. Again, we find that the feature size does not influence the performance too much. But based on the theoretical insights of learning minimal sufficient representation Achille & Soatto (2018), we believe that it is generally better to have a lower-dimensional representation while making sure it is sufficient for learning.

## E.2 ADDITIONAL EXPERIMENTAL RESULTS

**Ablation Study: Transferring Different Components**
Figure 5 shows the ablation study of our method in continuous control tasks. We compare our method with the following variants:
(1) learning auxiliary tasks without transfer,
(2) only transferring transition models $\hat{P}$ and
(3) only transferring reward models $\hat{R}$.

Compared with the single-task learning baseline (the blue curves), we find that all the variants of our method can make some improvements, which suggests that learning dynamics models as auxiliary tasks, transferring $\hat{P}$ and $\hat{R}$ are all effective designs for accelerating the target task learning. Finally, our method (the red curves) that combines the above components achieves the best performance, justifying the effectiveness of our transfer algorithm.

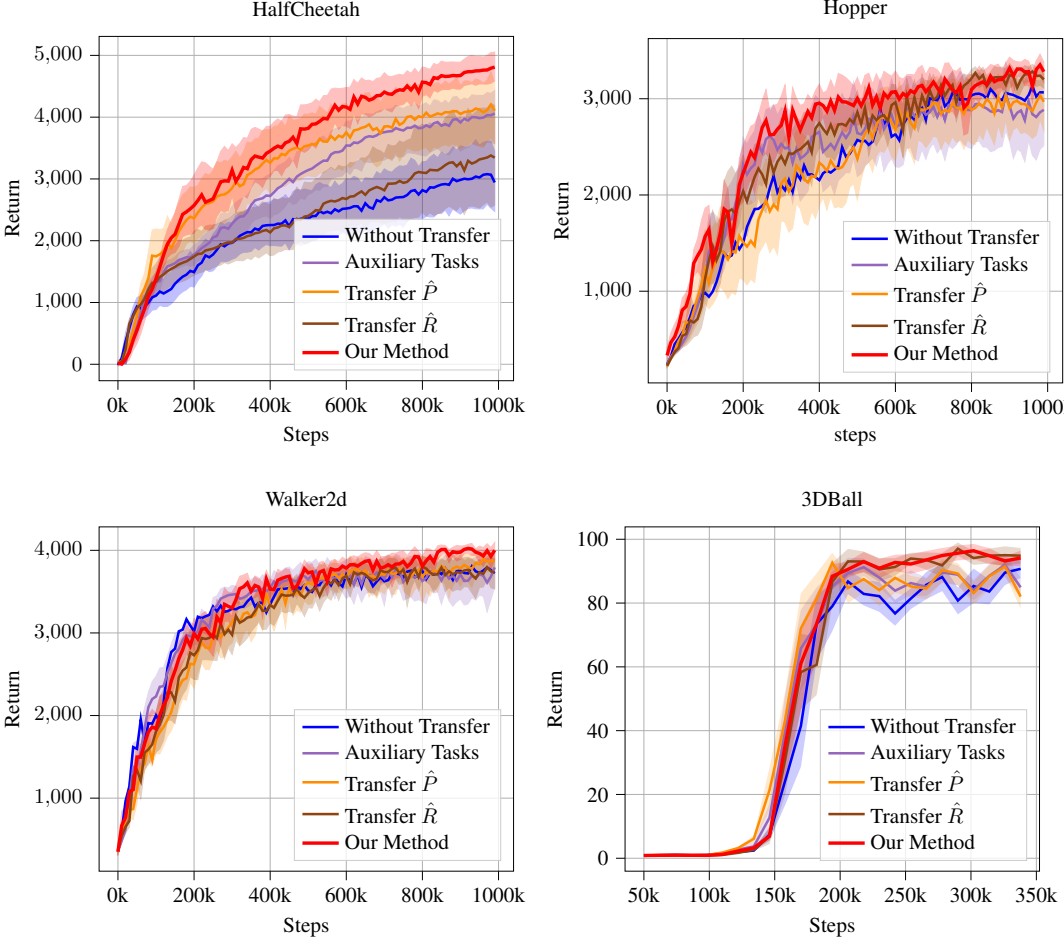

**Figure 5:** Ablation study of our method on different transferred components.

**Sanity Check: Effectiveness of the Proposed Transfer Method**

We conduct another ablation study to evaluate each component of our algorithm in the Cart-Pole environment as shown in Figure 6. We find that when transferring the dynamics models with only a linear value head (the **green** curve), the agent fails to learn a good policy as we analyzed in Section 4.3. If the dynamics models $(\hat{P}, \hat{R})$ are randomly generated instead of being transferred from the source task (the **orange** curve), the agent does not learn, either. More importantly, if we learn dynamics models as auxiliary tasks in the target task without transferring them from the source (the **purple** curve), the agent learns a little better than a vanilla agent, but is worse than our proposed transfer algorithm. These empirical results have verified our theoretical insights and shown the effectiveness of our algorithm design.

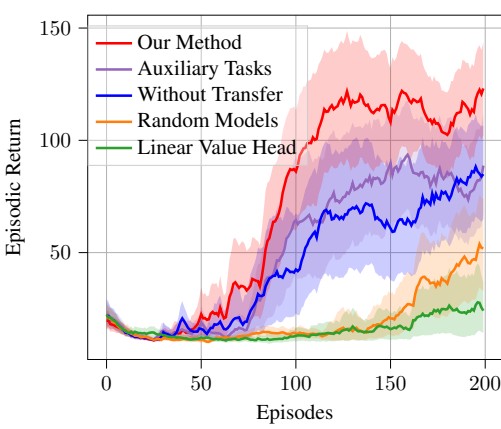

**Figure 6:** In the Vec-to-pixel CartPole environment, sanity check verifies the effectiveness of our algorithm design. Results are averaged over 20 random seeds.

**Hyper-parameter Test**

Figure 7 further visualizes how the hyperparameter $\lambda$ (regularization weight) influences the transfer performance in the Vec-to-pixel Cart-Pole environment. It can be found that the agent generally benefits from a larger $\lambda$, which suggests that the model-based regularization has a positive impact on the learning performance. For a wide range of $\lambda$'s, the agent always outperforms the learner without transfer (the learner with $\lambda = 0$). Therefore, our algorithm is not sensitive to the hyperparameter $\lambda$, and a larger $\lambda$ is preferred to get better performance. In Appendix E.1.3, we have provided the $\lambda$ selections for all experiments.

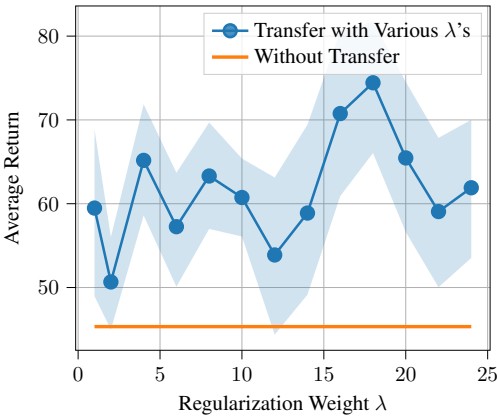

**Figure 7:** In the Vec-to-pixel CartPole environment, under different selections of hyperparameter $\lambda$, the algorithm works better than learning from scratch (when $\lambda = 0$). Results are averaged over 20 random seeds.

