# OpenReview forum: "Transfer RL across Observation Feature Spaces via Model-Based Regularization"
_ICLR.cc/2022/Conference — ICLR 2022 Poster_

### Official Review · Reviewer_igV9 · 2021-10-18

**Correctness:** 3
**Technical Novelty And Significance:** 2
**Empirical Novelty And Significance:** 2
**Recommendation:** 5
**Confidence:** 3

**Main Review:**

Overall, the paper is well-written, although some technical points are unclear. The problem that the authors consider is interesting and relatively novel. Specifically, the setting is close to many previously studied settings, but not identical to the best of my knowledge. The paper combines several theoretical results with empirical evaluation. There are three main issues with the paper: motivation, technical clarity, and experimental evaluation.

## Motivation

The setting the authors consider, transferring between observation spaces, has not been tried before to the best of my knowledge. The authors claim that it is an important problem, but I do not find the motivation for this very persuasive. The authors state that “Observation change is common in practice due to hardware upgrading, data restriction or curriculum design.”. Could specific citations be provided here? Figure 1 shows an example where observation spaces could change, but it seems to be an abstract toy example. It would be better to use a real-world example here. Maybe sim-to-real is a better example since simulated environments can expose data structures for the simulator, which real environments cannot.

Overall, I think the sim-to-real problem is the setup closest to what the authors are trying to achieve. I would encourage the authors to discuss sim-to-real more, and maybe use some baselines from there. See e.g. ​​Sim-to-Real: Learning Agile Locomotion For Quadruped Robots.


## Technical Clarity

The role of the approximation operator is a little unclear. What does this operator model? Does it just mean the best Q-value estimates available under given features? The paper would be easier to read if a motivating example was given when it was introduced.

Overall, the theoretical arguments appear to be somewhat circular. The authors define a representation mapping to be sufficient if it enables accurate Q-values. It is then shown that if a representation mapping is sufficient, one can obtain accurate q-values. I have a hard time assessing what the theoretical results really prove.

Is it possible to measure if some representation mapping is sufficient? If not, the definition might not be useful in practice.

Also, do the theoretical results apply to the empirical setting considered later?

For the experiment section, the setup details are relegated to Appendix D. There, however, there are not enough details to really understand what is going on. E.g., what is the size of the screen, and what is the size of the crop? What parameters, beyond learning rate, are used for adam? What settings are used for the replay buffer? For continuous control, are the actions discretized in any way to enable an argmax policy?

## Experimental evaluation

The authors state “We implement a DQN learner”. There are many high-quality DQN implementations available online, and doing your own implementation can introduce bugs. It is not clear that the DQN implementation works well, and if it does not, improving upon it is not very meaningful.

In Figure 4, results on tasks hopper and walker do not appear to be statistically significant.

How are the tasks selected? How are the architectures selected? How are the hyperparameters selected?

For latent space models, there are many excellent implementations available online e.g. dreamerV2. Why not use one of these?

Why are the proprioceptive states of the environments modified? E.g. in Appendix D, the authors write: “For the target task of MuJoCo environments, we add the center of the mass based inertia and velocity into the observations of the agent”.



## Post rebuttal update

I thank the authors for the many clarifications and improvements to the paper! I want to retain my score of 5 for two reasons. Firstly, I still think that the empirical results are not strong enough. In Figure 3, it seems like the improvements are only significantly outside the error bars for two tasks (Vec-to-pixel: Cheetah-Run and More-sensor: HalfCheetah). Secondly, while the authors have improved the motivation by giving relevant real-world scenarios where their methods apply, the experimental evaluation is pretty far away from these examples. Thus, while their methods improve performance for these toy tasks, I am not convinced their methods will improve a real-world problem. One specific direction to pursue here might be that of this paper (https://arxiv.org/abs/1912.12294).






**Summary Of The Paper:**

The paper considers the problem of RL environments where the observations space can change dramatically, e.g. from proprioceptive to pixel observations. The authors introduce various definitions for this setting and provide proofs regarding convergence (in the sense of Q-values converging to true values). Over 5 environments in continuous control, the authors demonstrate that their method enables transfer learning from proprioceptive to pixel observations and that such transfer learning can be beneficial.

**Summary Of The Review:**

While the paper often is well-written and considers an interesting problem, the motivation for the problem is lacking. Additionally, lack of technical clarity on both the empirical and theoretical sides makes it hard to assess what the paper really proves. At last, the empirical evaluation is problematic -- some results are not statistically meaningful, many parameter choices are not motivated and it is not clear that the baseline DQN implementation works well in the first place.

---

> ### Author Response · Authors · 2021-11-22
> **[Response 5/5] Summary and References**
>
> We again thank Reviewer igV9 for the feedback and suggestions on our paper. We hope that our revised paper and the answers above have addressed the concerns of the reviewer. Please let us know if there are further questions.
>
> ---
> Refs:
>
> [1] Nagabandi et al. Neural Network Dynamics for Model-Based Deep Reinforcement Learning with Model-Free Fine-Tuning. ICRA 2018.
>
> [2] Kurutach et al. Model-Ensemble Trust-Region Policy Optimization. ICLR 2018.
>
> [3] Buckman et al. Sample-Efficient Reinforcement Learning with Stochastic Ensemble Value Expansion. Neurips 2018.
>
> [4] Higgins et al. Darla: Improving zero-shot transfer in reinforcement learning.
>
> [5] Brockman et al. OpenAI Gym. 2016
>
> [6] Lai et al. On Effective Scheduling of Model-based Reinforcement Learning.

---

> ### Author Response · Authors · 2021-11-22
> **[Response 4/5] Clarifications on Experiment Setting and Implementation**
>
> > Q8: For latent space models, there are many excellent implementations available online e.g. dreamerV2. Why not use one of these?
>
> We do not use Dreamer for our latent space model because the purpose of our algorithm using the latent dynamics model is different from Dreamer. For our proposed algorithms, the only purpose of the latent dynamics model is to regularize the learned representation. Therefore we use a linear dynamics model in our experiments, and we find it works better than a deeper dynamics model. An intuitive explanation is that when the learned dynamics model is simple, the information of the state and action will be well-embedded in the learned representation. On the contrary, if the dynamics models are complex, the essential information may be fused and convoluted in the representation, making it hard to quickly find a good policy with the representation.
>
> So, as we explained at the end of Section 4.3 and Appendix B, we enforce the learned representation to fit the policy-independent dynamics, and we show that the resulted representation is sufficient for learning the target task. However, Dreamer uses the latent dynamics model to learn the control policy, and thus it is much more complicated than what we need in our algorithm. For this reason, we do not use the implementation of Dreamer.
>
> > Q9: How are the tasks selected? How are the architectures selected? How are the hyperparameters selected?
>
>
> We apologize if some details were missing in the previous manuscript. We have already updated the manuscript based on the suggestion (Section 6 and Appendix E.1). Below we address the proposed questions.
>
> **About task selection:**
>
> We would like to test our transfer algorithm on a wide variety of transfer learning settings satisfying Assumption 1 (Section 3), which mimics many applications discussed in the introduction.
> Therefore, we consider 3 different practical use-cases as described in Section 6: (1) From vector to pixel. (2) When there are more sensory inputs. (3) When there are broken sensors.
> For the above 3 scenarios, we choose 7 widely-used environments (we have added 2 more vector-to-pixel tests, Acrobot and Cheetah-Run).The properties of each environment are discussed in Appendix E.1.
> Therefore, we believe that our chosen environments are well-motivated, and they mimic many practical scenarios.
>
> **Network architecture and Hyper-parameters:**
>
> We have added more experiment setting details in Appendix E.1, including input format, base leaner architecture, hyperparameter, etc.
> In short, the architecture and hyperparameters of DQN and SAC are following existing codebases. The proposed transfer algorithm only requires one hyperparameter, $\lambda$, which controls the weight of the regularizer. Weight of regularization is a standard type of hyperparameters in many papers. Our selection of $\lambda$ for all experiments is listed in Appendix E.1.3.
> Importantly, in experiments, we find the performance does not change much with different $\lambda$'s. So we emphasize that the algorithm is not very sensitive to the value of $\lambda$, as shown in Appendix E.2.
>
> > Q10: Why are the proprioceptive states of the environments modified? E.g. in Appendix D, the authors write: “For the target task of MuJoCo environments, we add the center of the mass based inertia and velocity into the observations of the agent”.
>
> First, adding the mass-based inertia and velocity is a natural extension to the original observation space because these extra pieces of information are provided on the Gym Humanoid environment [5].
>
> Second, this modification emulates the real-world scenarios of robotics development, where the human developers constantly add more sensors or other features in order for the agent to learn a better policy.

---

> ### Author Response · Authors · 2021-11-22
> **[Response 3/5] Clarifications on Experiment Setting and Results**
>
> > Q5: For the experiment section, the setup details are relegated to Appendix D. There, however, there are not enough details to really understand what is going on. E.g., what is the size of the screen, and what is the size of the crop? What parameters, beyond learning rate, are used for adam? What settings are used for the replay buffer? For continuous control, are the actions discretized in any way to enable an argmax policy?
>
> We updated the manuscript based on Reviewer igV9's suggestions and discussed the experiment details in Appendix E.1.
>
> For the Vec-to-Pixel CartPole environment, the original screen size is (400,600,3), and we crop it to be (40,90,3) so that it is centered at the cart. In terms of the extra hyperparameters in the Adam optimizer, we use the default parameters from PyTorch, where $\beta_1=0.9$, $\beta_2=0.999$. For the replay buffer, we use a size of 10000.
>
> For the continuous control environments, we use SAC as the learner, which uses a Gaussian policy, as we mentioned in the paper. So there is no need to discretize the action.
>
> > Q6: The authors state “We implement a DQN learner”. There are many high-quality DQN implementations available online, and doing your own implementation can introduce bugs. It is not clear that the DQN implementation works well, and if it does not, improving upon it is not very meaningful.
>
> The reviewer might have misunderstood us as coding the DQN learner by ourselves. We would like to clarify that **we indeed used an existing DQN-for-Pixel-CartPole implementation,** as cited and discussed in Appendix E.1.
>
> In addition, our DQN implementation is mostly a proof of concept to show that the proposed transfer learning algorithms could be applied for efficient learning. So we do not include those advanced techniques such as Rainbow DQN or C51, which are independent of our investigation on the efficiency of the proposed algorithm. Please note that the proposed transfer method can be used for any based learning algorithm. Therefore, if more advanced base algorithms are used, the transfer method and the baseline method will both have improved performance.
>
> > Q7: In Figure 4, results on tasks hopper and walker do not appear to be statistically significant.
>
> First, the result on hopper has a significant improvement in terms of sample efficiency, since our transfer method converges much faster than a single-task learner. Please note that hopper is a relatively simple task, so that there is not much room for improvement in terms of asymptotic performance. The SOTA model-based method [6] also converges to ~3000 reward, which is similar to ours.
>
> Second, as we pointed out in the last paragraph of Section 6, this phenomenon also coincides with what is reported in several previous model-based RL papers [1,2,3], where the performance gain of their model-based approach is usually less significant on hopper and walker than on halfcheetah. In our opinion, the reason is mainly because learning dynamics models on hopper and walker is more challenging due to their relatively complex and non-smooth state shifting. We also observe that the learned latent dynamics models on these two specific environments are of relatively low quality (model loss is not low). This result verifies the correlation between model fitting and learning (Proposition 6 and Theorem 7). Therefore, as discussed in our paper, existing techniques of improving model-based learning can be used to improve the transfer performance.
>
> Moreover, we would also like to point out that even though the learned latent dynamics model is not very accurate, the proposed model-based regularizer still does not degrade the performance of the RL learner, which is a good property of our proposed algorithm.

---

> ### Author Response · Authors · 2021-11-22
> **[Response 2/5] Interpretations and Clarifications on Theoretical Results**
>
> > Q1: The role of the approximation operator is a little unclear. What does this operator model? Does it just mean the best Q-value estimates available under given features? The paper would be easier to read if a motivating example was given when it was introduced.
>
> Thank reviewer for the suggestion, we have updated the paper to make it more clear.
>
> The role of the approximation operator is just to model the process of fitting $Q^\pi$ using a specific hypothesis class $\mathcal{H}$, where $h\in\mathcal{H}$ is parameterized by $\theta\in\Theta$. In practice, $h$ could be a linear function or a neural network. Depending on the feature representation $\phi$, the approximator operator finds the optimal parameter $\theta^*$ in the hypothesis class so that the mean squared errors between the fitted Q values and the true Q values $Q^\pi$ are minimized.
>
> > Q2: Overall, the theoretical arguments appear to be somewhat circular. The authors define a representation mapping to be sufficient if it enables accurate Q-values. It is then shown that if a representation mapping is sufficient, one can obtain accurate q-values. I have a hard time assessing what the theoretical results really prove.
>
> In Definition 3, we define the sufficient condition of a representation mapping as the one that could fit the Q values of all policies in $\Pi_\phi^D$, given the hypothesis class $\mathcal{H}$. However, Definition 3 does not justify "how the sufficient representation helps with converging to a near-optimal policy via learning". Lemma 4 and Theorem 7 are exactly answering this question, where we showed that given a sufficient representation, we could have convergence guarantees in terms of policy iteration algorithms.
>
> > Q3: Is it possible to measure if some representation mapping is sufficient?  If not, the definition might not be useful in practice.
>
> The exact sufficiency is an ideal condition in theory that we would like the learned representation to have. We also provide an $\epsilon$-sufficient condition to include non-ideal cases, and both the Lemma 4 and Theorem 7 in our theoretical analysis take the $\epsilon$-sufficiency into consideration.
>
> Please note that the theoretical insight of sufficiency is the foundation of the designed algorithm, in which exact **measurement of sufficiency is not needed.** Instead, we utilize the results in Proposition 6, Theorem 7 and Theorem 8 to encourage the agent to learn a sufficient representation by learning and transferring an auxiliary latent dynamics model.
>
> Although it is not needed in our algorithm, if one does want to measure the sufficiency, our Definition 3 already provides a feasible solution: to measure whether the representation is able to fit the Q values of encoded policies sufficiently well.
>
> > Q4: Also, do the theoretical results apply to the empirical setting considered later?
>
> Our theoretical analysis sheds light on why transferring the dynamics model could help the representation on the target domain to be learned much more efficiently.
>
> In particular,
>
> (1) in Proposition 6 and Theorem 7, we show that if the representation is sufficient for predicting the dynamics, it is sufficient for learning. This is why learning an auxiliary dynamics model helps with learning.
>
> (2) in Theorem 8, we show that the dynamics model ($\hat{P}, \hat{R}$) learned in the source task are transferable to the target task, and there exists a sufficient representation $\phi^{(T)}$ on the target domain that matches ($\hat{P}, \hat{R}$). This is why transferring the dynamics model to the target task and fixing it enables sufficient and more efficient learning. (Efficient because it does not need to learn dynamics from scratch.)
>
> Therefore, the algorithm and the theory are closely related. Our proposed algorithm uses loss $L_{P}, L_{R}$ (Section 4.2) to enforce that the source-task agent learns a good latent dynamics model, and target-task agent learns a representation that fits this model, which are corresponding to Theorem 7 and Theorem 8, respectively.

---

> ### Author Response · Authors · 2021-11-22
> **[Response 1/5] Motivation and Real-world Applications of the Tackled Problem**
>
> We thank Reviewer igV9 for the valuable feedback. Below we address the proposed questions and concerns in detail.
>
> ### About Motivation
>
> We appreciate the advice of considering a sim-to-real examples. Although we did not discuss it in the previous introduction, we have some real-world examples in the ethics statement. Now we have added many motivating examples to the introduction.
>
> The transfer problem we consider is different from the common transfer settings, but we believe that it has many applications in practice. Below we describe some scenarios where our method can be applied.
>
> 1. **Game development.** RL is widely used to train NPC models in games. However, we know that the development of a game is incremental. There could be more and more scenes, characters, or obstacles added to the game world, and thus the observation space of the NPC agent would change (typically becomes larger and larger). But the game physics and reward distribution could remain unchanged. In this case, retraining a policy could be expensive, while our method can help improve the learning efficiency.
> 2. **Adaptation to a new observation distribution.** It is a well-known challenge to adapt to novel observations. A common assumption[4] in literature is that some samples from target-task observation distribution are available. However, this availability may be unrealistic in some cases. For example, one has trained a patrol robot in a room, but later on, the room is repaint/redecorated. The old policy is likely to fail since it did not learn from the new observation distribution. In this case, our method can be used to enable fast adaptation.
> 3. **Simulator to real.** Our Figure 1 was inspired by a sim-to-real problem. In a simulator, the absolute locations of objects can be hand-coded and passed to the agent. However, in the real world, it is hard to get the absolute locations, and the agent usually uses sensors/cameras to obtain observations. Therefore, knowledge transfer from a low-dimensional observation to a high-dimensional observation is useful.
> 4. **Curriculum design.** As mentioned in point 3 above, our method can transfer knowledge from low-dimensional vector observation to high-dimensional image observation. This can be naturally used for curriculum learning. Directly learning from image inputs may be hard, but we can first let the agent learn from an informative vector observation, then let it transfer the knowledge to image observations that will be encountered in the real world.
>
> Other than the above scenarios, observation change also happens when the data access and permission changes, or when the sensor number/type changes. Therefore, the problem we consider has many practical usages, but is rarely studied in the literature. This is why we would like to take the first step in this interesting direction. We have included these motivating real-world examples in our introduction.

---

> ### Author Response · Authors · 2021-11-29
> **Thank you for the update. We provide further clarifications on the experiment setting and results.**
>
> We thank Reviewer igV9 for reading our rebuttal and updating the review. We greatly appreciate the suggestions provided, especially the insightful related work.
>
> We would like to first emphasize that one of our contributions is to **identify this novel problem** of knowledge transfer between drastically different observation spaces, which has not been discussed in literature before. As we motivated in the introduction, this problem is important in many scenarios, and can inspire new lines of research in other areas such as curriculum learning. We believe that our work is an important first step towards solving this problem.
>
> > About empirical results.
>
> We have shown the error bars for all the environments, which suggests that other than Cheetah-Run and HalfCheetah, the proposed method is also outside of the error bars of baselines in CartPole, Acrobot and 3DBall. Please note that the learning curves of the proposed method are also more stable than baselines (less fluctuation, narrower error bars) due to a better representation regularized by transferred models.
>
> > About real-world scenarios
>
> We would like to point out that many papers in RL use real-world applications as the motivation, and conduct experiments in common testbeds in OpenAI gym environments. We believe that the gym environments, especially MuJoCo tasks are close to real-world applications, and are widely used in literature to justify the effectiveness of RL algorithms.
>
> As we summarize in page 8 (scenarios), the selected environment is designed to mimic real-world applications.
> - The Vec-to-pixel setting uses vector-based source task and pixel-based target task, which is exactly the example discussed in the 3rd paragraph in the introduction (curriculum learning via observation design).
> - The More-sensor setting adds more sensors to the agent so that the observation space is enlarged. This is corresponding to Example (1) and (2) in the introduction (incremental development of environment and sensor upgrade).
> - The Broken-sensor setting replaces the velocity sensor with a temporal stack of other sensors. This corresponds to Example (3) in the introduction (data restriction), since the velocity data is lost and replaced by other information to ensure Markovian properties.
>
> In addition, the observation space change in the 3DBall environment is an existing application in the ML-Agents toolbox, designed by game developers. Therefore, this environment is already a real-world problem.
>
> > About the related paper[1]
>
> We thank the reviewer for recommending an interesting paper [1] to us. We will add more discussion about it in our final version. We would like to emphasize that [1] further justifies the motivation and the advantages of our work, as detailed below.
>
> 1. [1] well motivates why it is useful to learn from ground-truth vector observations before learning pixel observations, which is also one of the motivations in our paper.
> 2. [1] requires a ground-truth map that maps pixel observations to ground-truth vectors. In contrast, we do not need such a mapping. (We only need to first learn from a vector-based environment, then learn a pixel-based environment. There is no need to build a mapping between two state spaces.) Note that in real-world applications, vector observations and pixel observations are collected by different sensors, but building a mapping between vectors and images usually requires extra work, e.g., human annotation, which can be expensive. Therefore, our method is practical and easy to use, without requiring any expert knowledge.
>
> ---
>
> We hope the above clarification can address your concerns. We are happy to discuss further. Thank you again for the inspiring comments!
>
>
>
> [1] Chen, Dian, et al. "Learning by cheating." Conference on Robot Learning. PMLR, 2020.

---

### Official Review · Reviewer_gWD8 · 2021-10-27

**Correctness:** 2
**Technical Novelty And Significance:** 2
**Empirical Novelty And Significance:** 2
**Recommendation:** 5
**Confidence:** 4

**Main Review:**

I generally like the idea that the paper follows and its theoretical rigorousness. However, I have a few concerns about the exposition of the paper as well as its empirical evidence. Also, I think the number of real-world applications this solution addresses is very limited, making it a very narrow field of application.

While I especially like the insights to non-linear policy heads at the end of Sec. 4, I suggest shortening Secs. 4.1 and 4.3 and reduce them to their general insights (putting the rest into the appendix). This makes more space available for the presentation and discussion of additional experimental results.


*** Motivation of the paper and real-world application. ***
At points I was confused by the motivation of the paper. This started in the very beginning with “the observation space is specified by human developers and restricted by physical realizations and may thus be subject to dramatic changes” – I wonder in which real-world use-cases this really is the case (I mean – without considering simulation environments etc.). Even more: switching from vector-based observation to image-based observation sounds interesting but might finally be a fictional problem setup.

A much more intuitive scenario would be a sensor fusion setup where you have different sensors that measure different physical properties but which, in their sum, are redundant. If you remove a sensor (breakage?) the policy becomes useless as it expects a different input encoding while still (as the authors propose) the dynamics remains the same.


*** Related work ***
-	The papers argues that it is not easy to scale up the work of Sun et al (2020) for continuous domains. However, while not originally being proposed it seems more or less straight-forward? This is also lined out in the original paper. Hence, this work might also serve as a baseline
-	The paper argues that the work proposed by Raiman et al (2019) does not work for drastic changes. I cannot completely agree. The originally proposed gradient mapping could also be applied to the embedding where there is a sufficient representation. This work could hence also serve as a baseline. Even if it fails for the ‘drastic change’ it might work on the proposed Mujoco experiments.


*** Experimental Results ***

Although the authors claim that there is no previous work to which they can compare their algorithm to, I have a few suggestions/remarks:
-	(see lsao before) In the introduction you mention previous work of Raiman et al. (2019) who add or remove individual observation features. This is exactly what the authors do in their Mujoco experiments. Hence, this work serves as a baseline to which the authors should compare their approach to
-	Also an option would be to not starting to train the policy from scratch but to fine-tune the policy on the new observation representation. At least for the Mujoco setup this should be a viable starting point.
-	As the authors effectively propose a model-based RL algorithm/extension they should also run their algos against MBRL baselines.

Questions on DQN & Cartpole:
-	What exactly has been used in DQN? Dueling Double DQN with prioritized experience replay and noise? What about exploration parameters? As the results are not clearly showing the approach to outperform I wonder how well it would perform under perfect DRL algorithm configuration….
-	Presenting a screenshot of the cartpole to the agent forms a POMDP as we do not get information on the angular velocity and the velocity of the pole’s tip – which are both the most important components. I wonder why the authors formulate the experiment like this. Or did you stack frames to ensure Markovian state representation?

Questions on SAC & Mujoco/3DBall:
-	In my opinion, the observation representation change in the Mujoco environments is too simplistic. There are only features stacked on top of the existing features. Here, I expect a fine tuning of the policy whily freezing anything but the early layers should outperform both approaches, right?
-	The 3DBall modification is much more challenging and effectively also resembles the motivation of the paper. However, the results are not interpreted at all? Moreover, the results seem not convincing. It looks like both perform en par while the “without-transfer” approach does not reach the final performance. (Or does it towards the end???) There is too few information on the task and environment to interpret the results….

In general, there is a lack of discussion of the results of the approach and its limitations. It would also be interesting to see an ablation study that shows how much the representation effectively needs to change so that the proposed approach beats a fine-tuning of the policy. Also: what is the motivation of using DQNs in the first experiment and SAC in the last? I have the feeling that in some/many combinations of DRL agents and environments the proposed method also might lack behind and retraining of the policy…

Minor remarks:
-	Typo in the introduction: automotive[-s-]
-	The concept of the target networks (p5 bottom) is confusing at first as it refers to target/evaluation nets known from DQN. At this point the readers first thinks that the paper talks about the network that encodes target environment dynamics.


**Summary Of The Paper:**

The paper addresses the problem of adapting a policy to a novel representation of the environment. Different from most previous work the paper focuses on the problem of a change in the representation of the observation provided by the environment, i.e., assuming that the underlying dynamics P and R remains mostly similar. The idea is to learn source and target models of the environment jointly with the policy. This serves as a model-based regularization that guides policy and representation towards the learned models. This effectively leads to a learned representation that is shared for the policy for the source and the target environment representation.

**Summary Of The Review:**

I generally like the idea of the paper but in my opinion the paper is not ready yet for publication in ICLR. The main reason for that is the lack of experimental evidence and lack of comparison to previous work exposed to ablated variants of the proposed problem setting. Moreover, the application of the methods seems quite narrow given the assumptions of drastic representation changes during evolvement of environments.

*** update of the review ***

I would like to thank the authors for their thorough submission update. The other reviews also share some of my criticism and the authors imho did a very good job updating their work. Exposition and motivation are much clearer know the the experiments have improved considerably.

Bottom line: an increase of my score is definitely justified and I will increase my score to 5. The reason while I am still not leaning towards acceptance is because I still have concerns:
The fine-tuning experiment is not what I actually meant. In the experiments only the policy head is kept but I suggested to keep on training with the new representation (on the cases where the observation just gets bigger; just add the missing neurons at the beginning and initialise them). This should be faster/better.
And against such a baseline, I am still missing an ablation study that answers the following question: How much different must a representation be such that the transfer approach is better?

P.S. there are two typos:
- feature is corresponding [to] which old observation feature.
- Page 8, last line: 'Fine-tine' instead of 'fine-tune'

---

> ### Author Response · Authors · 2021-11-22
> **[Response 4/4] Summary and References**
>
> We again thank Reviewer gWD8 for the feedback. We sincerely hope that our revised paper and the above clarifications have addressed your concerns. Please let us know if there are further questions.
>
> ---
> Refs:
>
> [1] Sun et al. Temple: Learning template of transitions for sample efficient multi-task rl.
>
> [2] Raiman et al. Neural network surgery with sets.
>
> [3] Higgins et al. Darla: Improving zero-shot trans-fer in reinforcement learning.
>
> [4] Gupta et al. Learning invariantfeature spaces to transfer skills with reinforcement learning.
>
> [5] Mnih et al. Human-level control through deep reinforcement learning.

---

> ### Author Response · Authors · 2021-11-22
> **[Response 3/4] Clarifications on Experimental Setting and Results**
>
> ### III. About Experimental Results
>
> **Baselines.**
>
> We thank Reviewer gWD8's suggestions on the selection of baselines. As we stated in the answer above, we have included the suggested fine-tuning method in our experiments. In addition, we have added another baseline, time-based alignment [4] explained in Section 6.
>
> Please note that what we propose is not a model-based RL algorithm that learns a model for planning, instead, the learned model serves as a regularizer for model-free learning. Therefore, the "auxiliary task" curves are already a baseline with a model-based regularizer but without transfer.
>
>
> **On DQN & Cartpole.**
>
> We use a standard DQN algorithm as proposed by Mnih et al.[5]. What we propose is a transfer mechanism that does not depend on the base learning algorithm. That is, if the base learner is a more powerful algorithm, the performance of transfer learning will also increase.
>
> For the screenshot, we use current_screen - last_screen as the observation, so the velocities of the cart and the pole can be obtained from the observation. We apologize for not explicitly explaining it in the paper. We have added it in the revised version. (Appendix E.1.2 describes all environments in detail.)
>
> In addition, we have added two more vector-to-pixel environments in Figure 3, which both demonstrate the advantages of our method.
>
> **On SAC & Mujoco/3DBall.**
>
> > the observation representation change in the Mujoco environments is too simplistic.
>
> As mentioned in [Response 2/4], the proposed Mujoco experiments also have a drastic change. The dimension of Hopper is changed from 11 to 91, and the dimension of HalfCheetach is changed from 17 to 145. Details are explained in Appendix E.1.2.
>
> > The 3DBall modification is much more challenging and effectively also resembles the motivation of the paper. However, the results are not interpreted at all? Moreover, the results seem not convincing.
>
> We have added more information and interpretations about 3DBall in the revised version (Appendix E.1.2).
>
> The performance of "without transfer" already converges within the trained steps. But since the curve is averaged over multiple seeds, and the learning is not stable (the performance fluctuates), the results clearly show that "with transfer" is better and more stable than "without transfer". Note that 100 is the highest possible score in this environment.
>
> We use the same default learning hyperparameters (e.g. batch size, learning rate) for both methods, as provided in the original codebase. So it suggests that transfer learning improves the performance of a single-task learner.
>
> **Concerns on Implementation Decisions.**
>
> > what is the motivation of using DQNs in the first experiment and SAC in the last? I have the feeling that in some/many combinations of DRL agents and environments the proposed method also might lack behind and retraining of the policy
>
> *For the selection of base learners:*
> We did not cherry-pick the combination based on the performance. The selection of algorithms only depends on the environment property and is standard. CartPole is a discrete-action environment, so DQN is suitable. For continuous control tasks like Hopper, DQN is not applicable, and SAC is a commonly used algorithm for these tasks. We added a paragraph discussing the base learners in Section 6, page 8.
>
> *For the "lack behind" concern:*
> In our experiment results, we find that the proposed method almost always improves the learning performance. Even when the model fitting is not very successful as in Walker2d, the transfer approach does not lack behind the policy in retraining. Our theoretical analysis also shows that the dynamics model learned in the source is transferable to the target task, suggesting that transfer should be helpful. In Figure 3, we included more experimental results covering multiple challenging scenarios. We also conduct ablation studies as in Figure 4, Figure 5 and Figure 6, which all verify the effectiveness of the proposed method.
>
> In summary, we have provided both theoretical and empirical justifications for why the method is effective. But if Reviewer gWD8 is still not convinced, we will appreciate it a lot if Reviewer gWD8 could share with us the reason why the reviewer feels the proposed algorithm will "lack behind" learning. With these detailed reasons, we can further conduct experiments to analyze and address the proposed concern.
>
> > Typo in the introduction: automotive[-s-]
>
> Thank Reviewer gWD8 for pointing it out. We have fixed it.
>
> > The concept of the target networks (p5 bottom) is confusing at first as it refers to target/evaluation nets known from DQN.
>
> We apologize for causing this confusion. We have replaced this name by "stable encoder".

---

> ### Author Response · Authors · 2021-11-22
> **[Response 2/4] Additional Baselines; Discussion on More Related Work**
>
> ### II. About Related Work
>
> We agree that there could be some baseline methods to compare with. We have updated our experiment section and **added 2 more baseline methods:** fine-tune and time-aligned, as described in the paper. The single-task learner and the learner which learns an auxiliary dynamics model are also valid baseline methods.
>
> We have seriously considered the two potential baselines suggested by Reviewer gWD8. But we found that they still do not work in our setting.
> - Although the algorithm proposed by Sun et al.[1] is for multi-task RL, its extension to continuous domains is not a transfer learning algorithm. Instead, they use the proposed method to learn a more accurate dynamics model based on the raw observation (instead of the representation), and the model is used for planning. In contrast, our method is a generic transfer approach that works with a model-free base learner (e.g. DQN). Although we also learn a dynamics model, this model is an auxiliary task for regularizing the representation, not used for planning.
> - Please note that the **proposed Mujoco experiments also have a drastic change,** as the observation dimensions in the new task are about 8$\times$ as many as that in the source (e.g. 11->91 for Hopper). More importantly, the Network surgery method[2] requires to build a mapping from each input feature to each parameter and to compute the difference between the source and the target. That is, [2] requires to know which input feature is changed to the new feature (an inter-task mapping). However, in our problem setup, the major challenge is the unavailability of the inter-task mapping. Therefore, even if [2] works in the drastic change on Mujoco, the comparison is not very fair to our method.

---

> ### Author Response · Authors · 2021-11-22
> **[Response 1/4] Motivation and More Applicable Scenarios of Our Work**
>
> We thank Reviewer gWD8 for the valuable feedback and constructive suggestions. We are encouraged that Reviewer gWD8 likes the idea and the theoretical rigorousness of our paper. We have improved the writing as Reviewer gWD8 suggested. Below we address the concerns of Reviewer gWD8 in detail.
>
> ### I. About Motivation
>
> The transfer problem we consider is different from the common transfer settings, but we believe that it has many applications in practice. We have added many motivating examples to the introduction. Below we describe more scenarios where our method can be applied.
>
> 1. **Game development.** RL is widely used to train NPC models in games. However, we know that the development of a game is incremental. There could be more and more scenes, characters, or obstacles added to the game world, and thus the observation space of the NPC agent would change (typically becomes larger and larger). But the game physics and reward distribution could remain unchanged. In this case, retraining a policy could be expensive, while our method can help improve the learning efficiency.
> 2. **Adaptation to a new observation distribution.** It is a well-known challenge to adapt to novel observations. A common assumption[4] in literature is that some samples from target-task observation distribution are available. However, this availability may be unrealistic in some cases. For example, one has trained a patrol robot in a room, but later on, the room is repaint/redecorated. The old policy is likely to fail since it did not learn from the new observation distribution. In this case, our method can be used to enable fast adaptation.
> 3. **Simulator to real.** Our Figure 1 was inspired by a sim-to-real problem. In a simulator, the absolute locations of objects can be hand-coded and passed to the agent. However, in the real world, it is hard to get the absolute locations, and the agent usually uses sensors/cameras to obtain observations. Therefore, knowledge transfer from a low-dimensional observation to a high-dimensional observation is useful.
> 4. **Curriculum design.** As mentioned in point 3 above, our method can transfer knowledge from low-dimensional vector observation to high-dimensional image observation. This can be naturally used for curriculum learning. Directly learning from image inputs may be hard, but we can first let the agent learn from an informative vector observation, then let it transfer the knowledge to image observations that will be encountered in the real world.
>
> Other than the above scenarios, observation change also happens when the data access and permission changes, or when the sensor number/type changes. Therefore, the problem we consider has many practical usages, but is rarely studied in the literature. This is why we would like to take the first step in this interesting direction.
>
>
> > A much more intuitive scenario would be a sensor fusion setup where you have different sensors that measure different physical properties but which, in their sum, are redundant. If you remove a sensor (breakage?) the policy becomes useless as it expects a different input encoding while still (as the authors propose) the dynamics remains the same.
>
> What Reviewer gWD8 mentioned is a good scenario where our method can be used. The 3DBall environment exactly matches this scenario. In 3DBall, we have a velocity sensor in the source task, and it is removed (broken) in the target task. However, as the velocity is not a redundant observation, one has to stack the past frames in the target task to infer the velocity, and thus the target observation becomes more challenging to learn. We can see from the results that the proposed transfer method achieves better performance and better stability than baselines. (Note that 100 is the highest score the agent can gain in this environment.)

---

> ### Author Response · Authors · 2021-11-30
> **Further Follow-up to Reviewer gWD8**
>
> Dear Reviewer gWD8,
>
> As we are approaching the end of the review discussion, we would like to kindly ask you to consider our revised paper and our previous response. We are very thankful to your suggestions and comments. As suggested, we have included many motivating examples in the introduction, added experiment results with more environments and more baselines, and explained all experiment setting details in the manuscript. Please let us know if there are any other questions.
>
> We again thank you for your time and consideration!
>
> Best regards,
>
> Papaer3216 Authors

---

### Official Review · Reviewer_L4Qx · 2021-11-02

**Correctness:** 3
**Technical Novelty And Significance:** 4
**Empirical Novelty And Significance:** 3
**Recommendation:** 8
**Confidence:** 3

**Main Review:**

## Pros
* The novel problem formation is well motivated from real-life applications.
* The structural similarity assumption 1 made for the problem setting is reasonable and the proposed solution elegantly solves the problem by leveraging the structural similarity.

## Cons/questions/suggestions
* __Comparisons that demonstrate the practical benefits when observation quality is improved are missing__ The example of upgrading sensors to improve observation quality is a great motivating example for the problem of transfer RL between different observation spaces. For some RL applications, by reducing the noise or augmenting the observations, one might be able to learn a better policy than the policy trained on the original representation. Based on this understanding, I think an experiment that demonstrates the usefulness of the proposed algorithm in this setting would greatly strengthen the current work. Specifically, one could study the transfer between pixel inputs or vector inputs with different levels of background noise or image resolution. The knowledge transferred from poorer observations should enable an efficient learning when the observation quality is improved (_which is shown in Fig. 4_), and the asymptotic performance of the policy under better representations should improve upon the poorer representation's corresponding policy (_this is missing_). If such results can be included, I believe readers can better appreciate the importance of this novel problem, because it might allow one to improve observation qualities to get better policies, without having to train from scratch on the new observations.
* __Details on observation encoder initialization__ For vector-image transfer, the encoder architectures would be different and the latent output $z$ could shift a lot for the same underlying state under the vector observation and the image observation. Would this large latent vector shift lead to unstable training? Does one need to control how these encoders are initialized?

## Minor comments
* The authors claim that the proposed method can make it easier to learn a good representation from the new observation space. The results in figure 3 and 4 indirectly support the authors' claim. I wonder whether it would be possible to directly compare the representation learning progress with/without the latent dynamics transfer. For example, use the learned representations for some auxiliary tasks such as clarification/prediction/clustering and compare the performance.


**Summary Of The Paper:**

This paper tackles the problem of observation space change for RL problems, i.e., the observation changes while the dynamics remain similar. To deal with this problem, the authors propose to learn a latent dynamics models based on encoded observations. When the observation space is changed, the learnt latent dynamics model can serve as a knowledge prior to transfer known dynamics information when learning with the new observation space.



**Summary Of The Review:**

The novel problem setting proposed by the authors is of practical interests and the authors's proposed solution is well justified. Though the proposed method does not provide significant benefit when it's hard to learn the transition dynamics, the novel problem formulation could inspire future work along this line to develop new methods that can achieve better transfer for more challenging environments.

---

> ### Author Response · Authors · 2021-11-22
> **Discussion on Practical Applications of Proposed Method**
>
> We greatly appreciate Reviewer L4Qx's positive feedback and insightful suggestions. We are particularly encouraged that Reviewer L4Qx finds the tackled problem novel and the proposed method elegant. Below we address the questions of Reviewer L4Qx in detail.
>
>
> > Q1: For some RL applications, by reducing the noise or augmenting the observations, one might be able to learn a better policy than the policy trained on the original representation.
>
> This is an interesting idea. Our main motivation is that the new observation may become more challenging due to some simulator-to-real gap, or missing observation dimensions, as motivated in our revised introduction. But Reviewer L4Qx is right that if the observation is made easier to learn, one can benefit from upgrading the observation, while not losing the dynamics information learned in the source task. We have added the source-task performance in Figure 3 for all environments, which shows that our transfer method does achieve **better asymptotic performance** with upgraded observations in Hopper, Walker2d and 3Dball. Although we could not run more experiments due to the time limit, we believe this is an interesting direction to explore. For example, one can use our proposed method for **curriculum learning.** In practice, observations are usually rich (e.g. images, languages), but directly learning from rich observation is challenging and expensive. Therefore, we can first let the agent observe the groundtruth information of the environment as a source task, then let the agent transfer the learned dynamics knowledge to ease learning in the richer real-world observations.
>
> We have updated the introduction section to include many real-world application scenarios of the proposed across-observation transfer learning, which can motivate our method better.
>
>
> > Q2: For vector-image transfer, the encoder architectures would be different and the latent output  could shift a lot for the same underlying state under the vector observation and the image observation. Would this large latent vector shift lead to unstable training? Does one need to control how these encoders are initialized?
>
> In our experiments, we randomly and independently initialize the encoders in both the source task and the target task. But we normalize the output of both encoders to be a unit vector (l2 norm is 1), which remedies the potential mismatch in their scales. We have added this detail in Appendix E.1.3.
>
>
> > Q3: I wonder whether it would be possible to directly compare the representation learning progress with/without the latent dynamics transfer. For example, use the learned representations for some auxiliary tasks such as clarification/prediction/clustering and compare the performance.
>
> This is an interesting point. The selection of the auxiliary tasks is important. The tasks used in supervised learning problems may not indicate the quality of the representation for an RL task. For example, if the representation can be used to classify the input images based on the contained objects, it does not necessarily suggest that the representation can learn a good policy, because the RL task may rely on different information of the image (e.g. object locations).
>
> In our paper, we believe that a good representation for RL should be able to represent different policy values, or to predict latent transitions and rewards. In this sense, our Figure 3 and Figure 4 have justified that the learned representation is good, as it helps with value fitting and policy learning.
>
>
> ---
>
> We again thank Reviewer L4Qx for the feedback and advice on our paper. We are happy to answer any further questions, or to discuss any possible extensions of our work.

---

> > ### Comment · Reviewer_L4Qx · 2021-11-29
> > **.**
> >
> > I'd like to thank the authors for their detailed response. The newly added application examples in the introduction section now better justifies the motivation of this work. I'd like to maintain my original recommendation of this manuscript.

---

> > > ### Author Response · Authors · 2021-11-29
> > > **Thank you for the recommendation of this paper**
> > >
> > > We thank Reviewer L4Qx for the response. We are encouraged that the reviewer thinks the motivation is better justified by the new examples. We believe this is a new research topic that has not been discussed much in literature. We hope this paper can inspire the study of new methods in this direction.

---

### Official Review · Reviewer_RpCP · 2021-11-03

**Correctness:** 3
**Technical Novelty And Significance:** 3
**Empirical Novelty And Significance:** 2
**Recommendation:** 6
**Confidence:** 3

**Main Review:**

Overall, I found this work interesting but slightly lacking in focus, and especially the theoretical parts are slightly divorced from the effective implementation. The empirical work is also quite similar to previous work, despite the rather different problem setting considered.

Here are additional comments and questions which would be worth clarifying:

1. The problem considered is quite novel and I feel could lead to novel applications and approaches. However, in my opinion this would be more interesting if only the environment dynamics were considered to be shared between the source and target task, and when action spaces would differ. In its current instantiation, this restricts this to very particular situations.
   1. A more generic solution would be to consider MDP homomorphism, something that was tackled by van der Pol et al, 2020.
   2. This paper isn’t cited and discussed and is quite clearly relevant to this current work, alongside more recent works from Amy Zhang.
   3. Some other works in bisimulation metrics and behavioral similarity might also deserve to be contrasted and discussed.
2. I liked how Section 4.1 explicitly spells out that a representation needs to support the policy improvement path and not just the fixed point. However, I am not yet fully convinced of how novel Definition 3 and Lemma 4 really are. The proofs appear quite similar in construction to many existing results in the (recent) bisimulation literature, and the end of this section makes statements which can be related to the Value Equivalence Principle from Grimm et al 2020. Similarly, the assumption of linearity brings the setting closer to successor features and their more novel versions, something that has been thoroughly explored over the years.
   1. Again, I’m looking to understand how novel and different the current work is compared to previous work which aren’t explicitly discussed in this draft.
3. The loss used in practice is not that related to the theoretical derivations, and hence the paper isn’t as focused as it could be.
   1. In effect, L_P and L_R are rather standard in the model-based literature (e.g. they are used exactly as is in Van der Pol 2020), so the only novelty is the constraint to still use the same frozen Transition and Reward models in the target task (which is interesting).
4. Figure 3 presents an interesting setting (transfering from joints to pixel observations), and the method does help.
   1. However, Figure a and b do not appear to have converged for the “Pixel Obs, Without Transfer” condition. I would not be comfortable drawing conclusions given the current curves.
   2. This is also performed on Cartpole, which is rather simplistic.
5. I wished Figure 4 was also covering a similar setting (joints -> pixels), instead of the rather arbitrary addition of extra joints.
   1. Results are also not as promising, no setting apart from HalfCheetah can convincingly demonstrate the benefit of the method.
6. I found Section D.2 in the Appendix quite interesting, especially the ablation about transferring P or R was something that I was about to ask about, so I wished it was more discussed. Recent work indicates that matching R in bisimulation metrics is prone to overfitting, so it’s interesting that these results do seem to indicate that it is still helpful to capture it.


References:
   * Van der Pol 2020: https://arxiv.org/abs/2002.11963
   * Zhang 2020, https://arxiv.org/abs/2003.06016
   * Castro 2020, https://arxiv.org/abs/1911.09291
   * Grimm 2020, https://arxiv.org/abs/2011.03506
   * Also slightly related: https://arxiv.org/abs/1911.12247 , https://arxiv.org/abs/2107.11676



**Summary Of The Paper:**

This paper proposes to leverage models of the environment and rewards as regularizers to perform explicit transfer between different observation spaces in RL.

Being able to quickly adapt to different observations is quite a different problem setting than what the rest of the literature is tackling (and we could argue on how relevant it is, but let’s leave that aside). They provide some theoretical analysis of this setting, especially one interesting result on when their method can converge to a near-optimal solution through approximate policy iteration.

They tackle it using ideas very similar to the recent wave of work leveraging bisimulation metrics, hence novelty / comparison on this aspect will be important.
They assess it on simple Mujoco control tasks, where their results are promising but perhaps still a bit early.

**Summary Of The Review:**

In conclusion, I found this paper interesting, and tackling a rather novel problem setting. However, it currently is quite theory centric albeit without contrasting the novelty of their analysis compared to previous works as much as I’d like it to.
In addition, the empirical instantiation is rather divorced from the analysis, and is not novel compared to the large body of literature assessing model-based regularization and representation learning.

---

> ### Author Response · Authors · 2021-11-22
> **[Responses 3/3] Summary and References**
>
> We again thank Reviewer RpCP for the detailed feedback and suggestions. We hope that our revised paper and the answers above have addressed the raised concerns. Please let us know if there are further questions.
>
>
> ---
> Refs:
>
> [1] van der Pol et al. Plannable approximations to MDP homomorphisms: Equivariance under actions.
>
> [2] Zhang et al. Invariant causal prediction for block mdps.
>
> [3] Castro, Pablo Samuel. Scalable methods for computing state similarity in deterministic markov decision processes.
>
> [4] Grimm et al. The value equivalence principle for model-based reinforcement learning.
>
> [5] Kipf et al. Contrastive Learning of Structured World Models.
>
> [6] Biza et al. The Impact of Negative Sampling on Contrastive Structured World Models.
>
> [7] Zhang et al. Learning invariant representations for reinforcement learning without reconstruction.
>
> [8] Lee et al. Stochastic latent actor-critic: Deep reinforcement learning with a latent variable model.
>
> [9] Barreto et al. Successor features for transfer in reinforcement learning.
>
> [10] Dabney et al. The value-improvement path: Towards better representations for reinforcement learning.
>
> [11] Nagabandi et al. Neural Network Dynamics for Model-Based Deep Reinforcement Learning with Model-Free Fine-Tuning. ICRA 2018.
>
> [12] Lai et al. On Effective Scheduling of Model-based Reinforcement Learning.

---

> ### Author Response · Authors · 2021-11-22
> **[Response 2/3] Clarifications on Theoretical Results and Experimental Results**
>
> > Q3: The loss used in practice is not that related to the theoretical derivations, and hence the paper isn’t as focused as it could be.
>
> The loss we use and the theoretical results are indeed closely related.  Proposition 6 suggests that if for all $o,a$, $\mathbb{E}_{o^\prime\sim P_a(o)}[\phi(o^\prime)] = \hat{P}_a(\phi(o))$, and $R_a(o) = \hat{R}_a(\phi(o))$ hold, then encoder $\phi$ is exactly sufficient for learning the task. The loss functions $L_P$ and $L_R$ are encouraging $\phi$ to satisfy the above conditions. If the losses are 0, then Proposition 6 guarantees the sufficiency of $\phi$. Moreover, if the losses are greater than 0, Theorem 7 shows that, as long as the maximum prediction error is small, the representation is still good for learning near-optimal policies.
>
> > Q4.1: Figure 3 presents an interesting setting. However, Figure a and b do not appear to have converged for the “Pixel Obs, Without Transfer” condition. I would not be comfortable drawing conclusions given the current curves.
>
> We have updated the paper with more steps for all methods in the CartPole environment. We find that in the pixel version of CartPole, the single-task learning method does not converge to a good policy even when it is given a lot more steps.
>
> > Q4.2: (Figure 3) This is also performed on Cartpole, which is rather simplistic.
>
> Now we have added new vector-to-pixel results for 2 more challenging environments: Acrobot and Cheetah-Run. They can be found in Figure 3 (first row).
>
> > Q5: I wished Figure 4 was also covering a similar setting (joints -> pixels), instead of the rather arbitrary addition of extra joints. Results are also not as promising, no setting apart from HalfCheetah can convincingly demonstrate the benefit of the method.
>
> We have **added the suggested experiments of joints -> pixels for MuJoCo** game Cheetah in Figure 3 (first row). But the purpose of Figure 4 (Figure 3 second row in the revised version) is to show more applicable scenarios of our proposed method. For example, a robot was explicitly given the velocity of an object, but later on, the velocity of this object is no longer available due to physical restrictions, and then the robot has to infer its velocity based on frame stacking. In this case, the observation change is drastic (joints -> stack of joints). We consider this scenario as an important application of our algorithm.
>
> About the improvement in tasks other than HalfCheetah: the results show that the **sample efficiency in hopper is significantly improved,** and the learning stability in 3DBall is also improved. Please note that 3DBall and Hopper are relatively simple tasks so that there is not much room for improving the final reward. The SOTA model-based method [12] also converges to ~3000 reward, which is similar to ours.
>
> As we point out in the last paragraph of the experiment section, the limited improvement in Walker2d is mainly due to the fact that fitting a dynamics model is relatively hard in Walker2d. A similar phenomenon is observed in other model-based works [11]. Therefore, we believe that techniques that can improve model-based learning can also be applied to further improve the transfer performance.
>
>
> > Q6: I found Section D.2 in the Appendix quite interesting, especially the ablation about transferring P or R was something that I was about to ask about, so I wished it was more discussed. Recent work indicates that matching R in bisimulation metrics is prone to overfitting, so it’s interesting that these results do seem to indicate that it is still helpful to capture it.
>
> We are glad to see that the reviewer finds our result interesting. We find that matching R is helpful, as it encourages the representation to distinguish states with different rewards. In general, we do not observe the issues of overfitting reward in our experiment. We think this is because reward prediction is only an auxiliary task in our framework, different from many bisimulation works that use the predicted reward [7] to measure the similarity among states.

---

> ### Author Response · Authors · 2021-11-22
> **[Response 1/3] Relation and Differences with Works Mentioned**
>
> We thank Reviewer RpCP for the valuable feedback and constructive suggestions. We are encouraged that Reviewer RpCP found the considered problem novel and our results interesting. Below we address the concerns of Reviewer RpCP in detail.
>
> > Q1: This would be more interesting if only the environment dynamics were considered ... very particular situations.
>
> We appreciate Reviewer RpCP's suggestion on considering different action spaces. Although we did not explicitly discuss the change in action space, our proposed method can be naturally extended to this case by learning an additional action encoder. But learning the action representation is out of the scope of this paper. We are happy to implement it in our future work.
>
> The recommended related works are all insightful and relevant. We have cited them and added more discussion in Section 4.2 and Section 5. Please note that our work is to learn an auxiliary dynamics model for improved representation learning, different from the model-based methods [1-6] whose focuses are to learn a better abstract model for planning. Leveraging our transfer method with these model-based methods could be an interesting direction. For example, the representation can be further regularized by a bisimulation loss. However, what we propose is a transfer learning approach across observation spaces, so the combination with existing representation learning methods is an extension of the current method and is not our main focus.
>
>
> > Q2: I am not yet fully convinced of how novel Definition 3 and Lemma 4 really are.
>
> Our theoretical analysis mainly addresses the following 3 questions:
> (1) What is a good representation/encoder for model-free RL?
> (2) How does learning an auxiliary dynamics model help with representation learning?
> (3) Is the auxiliary dynamics model transferable?
>
> The above 3 questions are answered by Definition 3, Theorem 7 and Theorem 8, respectively. We admit that Definition 3 and Lemma 4 are similar to some claims in prior works. However, Definition 3 and Lemma 4 mainly serve as foundations of our main results Theorem 7 and Theorem 8, which are different from prior works.
>
> To be more specific, our theoretical contribution is a systematical understanding of the relations among learning performance, representation quality, dynamics model fitting and cross-task transferability. Therefore, although there is a relation between our theoretical framework and existing results, our focus and results are significantly different from the literature. Below we analyze the differences with mentioned works in detail.
>
>
> **Difference with Bisimulation Literature**
>
> The bisimulation metric measures the similarity among states. It can be used to learn a representation that is invariant to task-irrelevant details [7]. But in our paper, we do not enforce the representation to match the bisimulation distance among states. We also do not use state reconstruction that is common in literature [8]. Therefore, our algorithm is relatively easier to implement and cheaper to use in practice.
>
> **Difference with Value Equivalence Principle**
>
> [4] establishes the relationship between policy values and models. Although we also build a relationship between value functions and models, our perspective and results are significantly different from [4]. We find that if the representation is sufficient for learning a policy-independent model, then the representation is sufficient for learning all important policy values. In contrast, [4] claims that models that yield the same policy values are equivalent, based on which [4] proposes a novel model-based learning algorithm. Note that we learn the immediate transitions and rewards for regularizing the representation in model-free learning, while [4] learns value-regularized models for model-based planning.
>
> **Difference with Successor Features**
>
> We first clarify that we do not have a linearity assumption. On the contrary, our analysis in 4.3 shows that the learned representation is not linearly sufficient for learning the RL task, and thus a non-linear policy head is used.
>
> The successor feature method [9] uses a linear decomposition of the action-value function, and learns generalizable features to enable fast adaptation in new tasks with the same state space but different reward. In contrast, our tackled new task has a different observation space and the same reward. As we discussed in Section 4.3, there is a trade-off between approximation complexity and representation complexity. If an encoder can fit the Q value with a linear transformation, the encoder has to be complex enough to capture the information of multiple policies [9,10], which is usually expensive to re-learn when observation changes.
>
> But we take a different and novel perspective: let the encoder fit a policy-independent dynamics model, and then the obtained encoder is sufficient for learning the task, and it does not need to recover the values via a linear transformation.

---

> ### Author Response · Authors · 2021-11-30
> **Further Follow-up to Reviewer RpCP**
>
> Dear Reviewer RpCP,
>
> As we are approaching the end of the review discussion, we would like to kindly ask you to consider our revised paper and our rebuttal. We appreciate your suggestions and comments a lot. Our previous response has addressed the concern about our novelty in detail. As suggested, we have included many motivating examples in the introduction, added experiment results with more environments and more baselines, and discussed the comparison to recommended related works. Please let us know if there are any other questions.
>
> We again thank you for your time and consideration!
>
> Best regards,
>
> Papaer3216 Authors

---

> > ### Comment · Reviewer_RpCP · 2021-12-01
> > **Thank you for your response**
> >
> > Sorry for the delay in responding, and thank you for all your work in improving the paper during this period.
> >
> > 1. The comparison to related work is very valuable, and I think this makes the contributions of this work much clearer now. The improved Introduction and Related Work are very appreciated.
> > 2. Sorry about my confusion about the linearity assumption. The fact that you started with a linear approximation example in Section 4.1 made me believe you were using this throughout. As you mention 4.3 does explicitly mention when this shouldn't apply. It might be helpful to reword that particular sentence to avoid confusing others ("For example, if a linear approximation ...". I would introduce H_\phi first and only then talk about a linear or non-linear formulation of it.)
> > 3. The additional results, and especially the reworked Section 6 and Figure 3 provide great improvements to the paper, and are very interesting to see. As discussed by the authors, it is still the case that some settings of Figure 3 do not show the clearest benefit of their method, but overall this is now strong enough in my opinion.
> >
> > Hence I'm increasing my score to indicate this improvement and would now lean towards acceptance.

---

> > > ### Author Response · Authors · 2021-12-01
> > > **Thank you!**
> > >
> > > We are very thankful for your response and the updated rating. We are encouraged that you think the updated introduction, related work, and experiment are stronger than before.
> > >
> > > For the linearity discussion, we are sorry if the example in Section 4.1 is misleading. We will change it in our final version. Thank you for the valuable advice!

---

### Author Response · Authors · 2021-11-22
**Response to All Reviewers: Summary of Paper Revision**

We thank all reviewers for their insightful questions and valuable feedback. We are encouraged that reviewers consider the tackled problem novel (*RpCP, L4Qx, igV9*), the theoretical results interesting (*RpCP*) and rigorous (*gWD8*), and the proposed solution elegant (*L4Qx*).

We have addressed individual questions of reviewers in separate responses. In the revised version, we incorporated all reviewers' suggestions by adding more motivating examples, more comparisons to prior works, more experimental results and baselines, as well as more implementation details. Here we briefly outline the updates to the revised submission for the reference of reviewers.

### Paper Updates:

- **[Section 1 - Introduction]** We added several real-life scenarios to motivate the tackled across-observation transfer problem. (*RpCP, gWD8, igV9*)
- **[Section 2 - Preliminary]** We removed some basic RL preliminaries to Appendix A to make this section shorter and more focused.
- **[Section 4 - Methodology]** (1) We added more interpretation on the theoretical results (*RpCP,igV9*) (2) We explained the approximation operator more clearly in Section 4.1. (*igV9*) (3) We changed the name of the target encoder to avoid confusion. (*gWD8*) (4) We explained the difference and relation between the proposed method and related works in Section 4.2. (*RpCP*) (5) We shortened the discussion about the non-linear policy head. (*gWD8*)
- **[Section 5 - Related Work]** We cited and discussed more related papers. (*RpCP*)
- **[Section 6 - Experiments]** (1) We added 2 more baselines: fine-tuning with the previous policy head, and time-based alignment[1] which explicitly aligns the source and target representations. (*gWD8*) (2) We added 2 more vector-to-pixel environments (Acrobot and Cheetah-Run), and train more steps for CartPole. (*RpCP*) (3) We re-organized the results to demonstrate our transfer performance in 3 different scenarios, matching 3 different real-world applications. (4) We moved some ablation test results to the main paper. (*RpCP*)
- **[Appendix E]** (1) We added more detailed explanations of the environments, the baselines, and the base RL algorithms. (*gWD8, igV9*) (2) We moved the hyperparameter test experiment results and some ablation results to Appendix E.2. due to the space limit.

We again greatly appreciate all reviewers' suggestions. We hope that our paper updates and responses have addressed reviewers' questions and concerns. Please let us know if there are further questions.

---
Refs:

[1] Gupta et al. Learning invariant feature spaces to transfer skills with reinforcement learning.

---

### Decision · Program_Chairs · 2022-01-20

**Decision:**

Accept (Poster)

**Comment:**

This work suggests using models of the environment as regularizers for performing explicit transfer in RL. Here are some of the highlights from the reviews and subsequent discussions:
  * Novel problem
  * Unclear to some of the reviewers why the problem setting is in fact important.
  * Well-written
  * Interesting theoretical results
  * Somewhat limited experimental results
Post-rebuttal, while there is not necessarily a great consensus, the reviewers all feel that it's an improved piece of work. While I am myself not fully convinced that the problem setting motivation truly aligns with the kind of empirical results that the work provides, on the balance I think this work is interesting and has sufficient novel contributions to be accepted at ICLR.